# Unraveling the glycosylated immunopeptidome with HLA-Glyco

Georges Bedran [1,2,7], Daniel A. Polasky [2,7], Yi Hsiao [3], Fengchao Yu [2], Felipe da Veiga Leprevost [2], Javier A. Alfaro [1,4,5], Marcin Cieslik [2,3,6] & Alexey I. Nesvizhskii [2,3] ✉

Recent interest in targeted therapies has been sparked by the study of MHC-associated peptides (MAPs) that undergo post-translational modifications (PTMs), particularly glycosylation. In this study, we introduce a fast computational workflow that merges the MSFragger-Glyco search algorithm with a false discovery rate control for glycopeptide analysis from mass spectrometry-based immunopeptidome data. By analyzing eight large-scale publicly available studies, we find that glycosylated MAPs are predominantly presented by MHC class II. Here, we present HLA-Glyco, a comprehensive resource containing over 3,400 human leukocyte antigen (HLA) class II N-glycopeptides from 1,049 distinct protein glycosylation sites. This resource provides valuable insights, including high levels of truncated glycans, conserved HLA-binding cores, and differences in glycosylation positional specificity between HLA allele groups. We integrate the workflow within the FragPipe computational platform and provide HLA-Glyco as a free web resource. Overall, our work provides a valuable tool and resource to aid the nascent field of glyco-immunopeptidomics.

Protein glycosylation has been extensively studied and found to play a variety of biological roles, including antigen recognition, host-pathogen interactions, and immune modulation[1]. Glycosylation causes dramatic alterations in response to cancer and has been suggested as a potential biomarker[2–5]. Moreover, glycosylation could be an attractive source of tumor-specific antigens, considering the viability of post-translational modifications (PTMs) on MHC-associated peptides[6–9] (MAPs). Critically, glycosylation has been reported to have a significant impact on the immunogenic properties of MAPs in terms of T-cell recognition[10–12] and epitope generation, owing to interference with proteolytic cleavage[13].

High-throughput identification of glycosylated MAPs from mass spectrometry (MS) data involves a combination of two notoriously challenging problems in computational proteomics. First, the proteolytic processing of MAPs requires nonenzymatic searches (i.e., non-

specific cleavage of proteins at every peptide bond). Considering all possible cleavages of reference proteins results in an enormous search space for candidate sequences. Second, the non-templated nature of the glycosylation process results in hundreds of distinct glycans that can be detected across the proteome[14]. A combinatorial explosion occurs when all possible nonenzymatic peptide sequences with many possible glycans are considered. As a result, a non-specific glycopeptide search is not feasible with many search engines due to prohibitively long run times and/or insufficient sensitivity. Despite recent advancements in glycoproteomics search engines with adequate processing speed, they do not possess the capability to conduct non-specific searches and are not suited for glyco-immunopeptidomics analyses[15–17]. To the best of our knowledge, very few glycosylation analyses have been performed on MAPs. One of the earliest successful identifications of glycosylated class II MAPs was made in 2005[18], with

¹International Centre for Cancer Vaccine Science, University of Gdansk, Gdansk, Poland. ²Department of Pathology, University of Michigan, Ann Arbor, MI, USA. ³Department of Computational Medicine and Bioinformatics, Ann Arbor, MI, USA. ⁴Department of Biochemistry and Microbiology, University of Victoria, Victoria, BC, Canada. ⁵School of Informatics, University of Edinburgh, Edinburgh, UK. ⁶Michigan Center for Translational Pathology, University of Michigan School of Medicine, Ann Arbor, MI, USA. ⁷These authors contributed equally: Georges Bedran, Daniel A. Polasky. ✉e-mail: nesvi@med.umich.edu

two N-linked glycopeptides found in an EBV-transformed human B-lymphoblastoid cell line. In 2017, Malaker et al. successfully identified 26 glycosites in three melanoma cell lines[9]. Both studies required identification of glycopeptides by manual annotation of the spectra. More recently, a third effort from 2021 captured 209 unique human leukocyte antigen (HLA) II-bound peptide sequences from the SARS-CoV-2 virus[19] using an automated glycopeptide search method assisted with manual verification of all glycopeptide spectra.

The large-scale analysis of glycosylated MHC-associated peptides (MAPs) presents significant challenges, primarily due to the enormous search space of glycosylated nonenzymatic peptides. To address these challenges, our recent developments in improving search speed[20] (MSFragger) and addressing the complexity of glycosylation[21] (MSFragger-Glyco) have proven to be beneficial. However, it is worth noting that currently, glycosylated MAPs are queried from data not enriched for glycosylated peptides, which poses a challenge as available false discovery rate (FDR) strategies are not suitable for this type of data. In light of this, we leveraged these advances to optimize a workflow for HLA-glyco searches and developed an improved FDR strategy specifically for glycosylated MAPs obtained from non-enriched MS-based immunopeptidome data.

We evaluated several FDR strategies, and implemented a group-specific glycopeptide FDR approach that filters glycopeptides and regular peptides separately, allowing for different cut-off scores in each group. To further refine the results and ensure the accuracy of the identified glycans, we applied additional glycan-level FDR filtering (glycan $q$-value threshold of 0.05) after peptide-level filtering to remove glycoPSMs without strong spectral evidence for the glycan. We assembled, carefully annotated, and analyzed eight publicly available immunopeptidomic datasets for N-glycosylation using our workflow and investigated the glycosylated MAPs binding properties. From nearly 2000 LC–MS/MS runs, we found 3409 class II N-glycosylated MAPs on 1049 distinct protein glycosylation sites of 677 unique proteins. We revealed characteristics of HLA glycopeptides, including high levels of truncated glycans, conserved HLA-binding cores across the 72 studied HLA class II alleles, and a different glycosylation positional specificity between the classical allele groups.

Our study advances our knowledge of glyco-MAPs in cancer, and with our optimized computational workflow represents a valuable resource for the growth of the field of glyco-immunopeptidomics. The role of MHC class II in inducing expression and antigen presentation on tumor cells is gaining recognition as a crucial mediator in promoting antitumor immunity and neoantigen efficacy[22–27]. Therefore, our study may contribute to a better understanding of the role of MHC class II in promoting antitumor immunity and neoantigen efficacy through glyco-MAPs. Our results are conveniently available as a free Web resource with the following link: https://hla-glyco.nesvilab.org.

## Results

### Computational glyco-immunopeptidomics workflow

The computational workflow developed in this study for the analysis of glycosylated MAPs is shown in Fig. 1. While O-glycosylated MAPs are also of potential interest[28], O-glycopeptide analysis typically requires electron-based activation to locate glycosite(s) within the peptide. As the vast majority of available immunopeptidomics data lacks such activation, we focused exclusively on N-glycosylated MAPs for this analysis. Our approach adapts the MSFragger-Glyco workflow to address immunopeptidomic searches that lack enzymatic digestion and glycosylation enrichment. N-glycans are covalently attached to peptides at a highly conserved site, known as a consensus motif or sequon, which follows the N-X-S/T pattern. Moreover, glycopeptides produce distinctive fragment ions in tandem mass spectra (i.e., oxonium ions), which is indicative of their presence. To identify a glycopeptide from a given spectrum, MSFragger-Glyco imposes strict requirements, including the presence of these characteristic oxonium ions in the spectrum as well as the occurrence of a sequon within the peptide sequence. By employing these stringent criteria, MSFragger Glyco can accurately and reliably distinguish glycopeptides from other peptide species, thereby enabling a comprehensive and high-throughput analysis of protein glycosylation.

Glycopeptides typically comprise a sizable portion of the acquired spectra in the context of glycopeptide enrichment, allowing the application of a single FDR strategy for all spectra. The previously published MSFragger-Glyco workflow, which is tailored for large-scale glycoproteomics experiments, applies a target-decoy approach in which reversed N-glycan sequons are verified in reversed (decoy) peptides to guarantee an equal proportion of possible glycopeptides in the target and decoy search spaces. We have demonstrated[21] that this approach can accurately and efficiently identify glycopeptides from the glycopeptide enriched data. Despite the significant benefits of glycopeptide enrichment, it is not yet feasible for MHC-associated peptide analysis owing to technical challenges that have yet to be

**The HLA-Glyco workflow**

**Fig. 1 | The HLA-Glyco workflow for the detection of glycosylated MHC-associated peptides.** The FragPipe suite was used to (I) perform a search for glycosylated peptides (glyco search) with the MSFragger search engine; (II) control the FDR with PeptideProphet in combination with a modified version of Philosopher; and (III) assign a glycan composition for each glycopeptide-spectrum match using PTM-shepherd.

overcome. This is mostly due to the large amounts of starting material required for the isolation protocol in immunopeptidomics experiments. In this case, the identification of glycopeptides becomes even more challenging, particularly in nonenzymatic searches, as glycopeptides represent a minority of acquired spectra.

To address this challenge, we first assessed the effectiveness of the standard false discovery rate (FDR) procedure described above when applied to non-enzymatic, unenriched immune glycopeptides. We observed that 91% of the glycosylated PSMs matched known glycosylation sites, and less than half (46%) of the glycosylation sites were previously reported by proteomic analyses (Supplementary Fig. 1a). The identified glycosylation sites with fewer supporting spectra and lower scores were mostly unknown (i.e., unreported by proteomic analyses), indicating a higher false discovery rate. As the presence of glycosylated PSMs in the identified spectra is relatively low (<5%), the score thresholds used in standard FDR filtering are primarily influenced by non-glycosylated peptides. This leads to an enrichment of false discoveries in the glycosylated fraction when all PSMs are filtered together due to the much larger search space of the non-glycosylated fraction with unenriched data. To improve identification quality, we applied a modified FDR approach using a separate PeptideProphet probability filter for glycosylated and non-glycosylated PSMs with Philosopher software. As a result, the proportion of glycosylated PSMs from known glycosylation sites rose significantly to 95.8%, while the proportion of unique known glycosylation sites also increased to 65% (Supplementary Fig. 1b). Further improvements were made by filtering glycoPSMs using a glycan $q$ value of $q \leq 0.05$ provided by PTM-Shepherd[29], which removed glycopeptides without sufficient evidence of glycan composition assignment[30]. Using this FDR method, the proportion of PSMs corresponding to known glycosylation sites increased to 96%. Moreover, the proportion of identified glycosites corresponding to known glycoproteins increased to 95%, with 79% of the sites previously identified in other glycoproteomic analyses (Supplementary Fig. 1c). In addition to our primary analysis, we conducted an entrapment search (i.e., combined database of human and *Arabidopsis thaliana* proteins) to further evaluate the effectiveness of our false discovery rate (FDR) strategy. Our FDR filtering yielded no glycoPSMs associated with *A. thaliana* (Supplementary Note 2 and Supplementary Data 1) highlighting the ability to accurately distinguish true glycoPSMs from noise. Hence, our stringent glycopeptide-specific filters provided effective FDR control in a challenging search and allowed for confident construction of the HLA glycopeptide resource.

## Large multi-tissue MHC immunopeptidome dataset

We selected eight immunopeptidomic studies[31–38], prioritizing studies with a large amount of high-resolution mass spectrometry data and included a variety of instruments as a means to reduce instrument bias (Supplementary Data 2, Supplementary Note 1, and "Methods"). Based on our careful curation and annotation of these data, our collection of 732 different HLA class II mass spectrometry samples incorporated 90.8% of HLA-typed data (Fig. 2a), 80.3% of patient tissues, 16.7% of cell lines, and 2.9% of tumor-infiltrating lymphocytes (Fig. 2b). The aforementioned samples covered up to six different cancers (Fig. 2c) located in the brain (meningioma and glioblastoma), skin (melanoma), colon (colorectal), and lung (adenocarcinoma and squamous carcinoma). In addition, 59% of the samples were non-cancerous and originated from disease-free individuals. In terms of HLA diversity, up to 72 HLA class II alleles of the three classic genes (DP, DQ, and DR) were covered by varying numbers of mass spectrometry samples (Fig. 2d).

Leveraging the wealth of proteomic data, we queried the glycosites identified in our study against previously reported glycosylation sites in GlyGen[39]. PSM level information showed 96.4% of previously reported glycosylation sites (Fig. 2e), 1.8% of glycosylation sites within previously reported glycosylated proteins, and 1.8% of new glycosylation sites. On the other hand, at the peptide level, 90% of

glycopeptides mapped to previously reported glycosylation sites, 6.7% of glycopeptides were within previously reported glycosylated proteins, and 3.3% contained new glycosylation sites. A similar trend was observed at the glycosylation site level, with 78.8% of previously reported glycosylation sites, 15.6% of glycosylation sites within previously reported glycosylated proteins, and 5.5% of new glycosylation sites. It appears that peptides containing previously reported glycosylation sites are abundant species, considering the high spectral count (Fig. 2f in gray) in comparison with the previously unreported ones (Fig. 2f in blue and black). We then benchmarked our findings against previous work by Malaker et al.[9] on glycosylated MAPs in three melanoma and one EBV-transformed B-cell line. The original manuscript reported 93 glycosylated peptides corresponding to 26 glycosylation sites, split between N-glycosylation (23) and O-glycosylation (3). Our workflow recovered 20 of the 23 identified N-glycosylation sites, of which 4 did not pass the FDR filter. With a 45-fold increase in glycosylation sites, we identified 1033 new sites (see Fig. 2g and Supplementary Data 3).

Overall, 78.8% of the detected glycosylation sites were consistent with those reported in GlyGen in previous glycoproteomics studies. Of the remaining 21%, 15.6% were detected in well-known glycoproteins, whereas only 59 were found in proteins that were not previously known to be glycosylated in GlyGen. This disparity is anticipated due to the variations in the workflows used in glycoproteomics and immunoproteomics, which target distinct parts of the proteome. Despite this, the high concurrence with previously annotated sites (80%) and known glycoproteins (95%) suggests that the identified glycosylation sites are reliable.

## Enrichment of N-glycosylation in the class II immunopeptidome

Several of the datasets we searched contained both HLA class I and II peptides from the same samples and, in one case, whole proteome data, allowing us to compare the frequency and characteristics of glycosylation across these categories. Fragmentation of glycopeptides by tandem MS (MS/MS) produces highly abundant oxonium ions resulting from the fragmentation of conjugated glycan(s), which can provide an estimate of the fraction of glycopeptides in a sample prior to a database search. To understand the abundance of glycosylation at different molecular levels, we compared the percentage of oxonium-containing MS/MS scans for four datasets containing multiple HLA classes (Fig. 3a). Interestingly, datasets A[31], B[34], and D[37] showed, on average, an approximate 5-fold enrichment in potential HLA class II glycosylation events compared with HLA class I data. In dataset C[32], the only dataset containing samples derived from healthy tissues, a similar proportion of oxonium-containing scans was observed in the HLA class II data as in the other datasets, but there were essentially no oxonium-containing scans in the HLA class I data. The absence of glycosylated HLA class I MAPs in the healthy samples of dataset C is intriguing. Nonetheless, it is worth considering that the data was obtained using a very restricted precursor m/z range, which could have led to the exclusion of glycopeptides from the analysis due to their higher molecular weight. As expected, the percentage of glycosylated PSMs obtained from database searches of these datasets followed a similar trend, with 0.5 to 3% of observed PSMs glycosylated in HLA class II data versus less than 0.1% glycosylated in HLA class I data (datasets A, B, and C). Strikingly, glycosylated PSMs were also enriched approximately 7-fold in HLA class II compared with the whole proteome data in dataset D (Fig. 3b), a dramatic increase given the abundance of glycosylation in the proteome. The increased level of glycosylation observed in HLA class II MAPs may be attributed to their origin from the extracellular proteome, which is expected to be more enriched in N-glycosylation[40] compared to corresponding whole proteome (cell lysate) samples.

We also noticed that the composition of glycans observed in the immunopeptidomic datasets was different from that of their

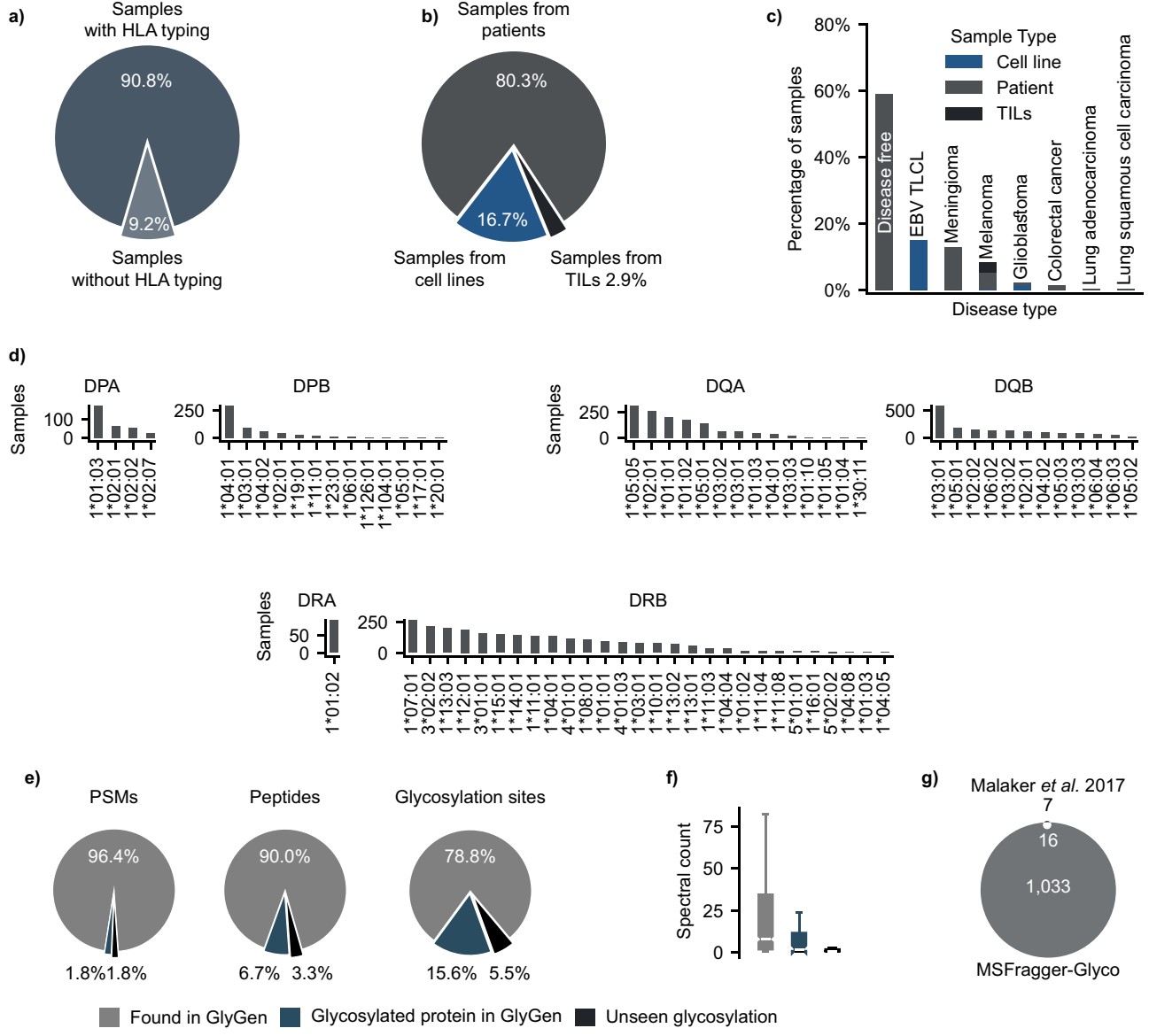

**Fig. 2 | HLA class II infographics of the eight datasets collected in this study.**
**a** Percentage of samples with HLA class II typing information. **b** Types of the collected mass spectrometry samples (i.e., patient tissues, cell lines, and tumor-infiltrating lymphocytes/TILs). **c** Cancer types across the collected mass spectrometry samples. **d** HLA class II alleles (DR, DB, and DQ) across the collected mass spectrometry samples. **e** Percentage of glycoPSMs, glycopeptides, and glycosylation sites found in GlyGen. **f** Abundance of the three categories from panel (**a**) by spectral count of n = 106 biologically independent samples. Gray boxplot values: Bottom whisker: 1, Q1: 1, Median: 1, Q3: 5, Interquartile range: 4, Top whisker: 11, Minimum: 1, Maximum: 13700. Blue boxplot values: Bottom whisker: 1, Q1: 1, Median: 1, Q3: 2, Interquartile range: 1, Top whisker: 3, Minimum: 1, Maximum: 335. Black boxplot values: Bottom whisker: 1, Q1: 1, Median: 1, Q3: 2, Interquartile range: 1, Top whisker: 3, Minimum: 1, Maximum: 501. **g** Comparison of the identified glycosylation sites with those reported by Malaker et al.[9].

proteome counterparts. (Fig. 3c). The average glycan mass detected in the immunopeptidome of dataset D was approximately 1000 Da, which is significantly lower than that observed in the corresponding proteome (1400 Da average). To further explore the nature of this compositional discrepancy, we compared the glycan types between the two groups (Fig. 3d). A higher percentage of truncated glycans (68%) was observed in the HLA class II immunopeptidome than in the more typical high-mannose and complex/hybrid categories in the proteome, as noted in a previous analysis[9]. This trend in truncated glycans on HLA peptides was preserved when only glycans from the same protein were considered. For example, LRP1, a highly glycosylated protein, was observed with a mix of high-mannose and complex glycans in the proteome sample, but with a mix of truncated and high-mannose glycans in the HLA-II sample with almost no mature complex glycans detected (Fig. 3e). There was little overlap between the

glycosylated proteins and sites in each category, with only 22.8% of HLA-II glycoproteins observed in the whole proteome data, and even lower overlap (16.3%) when considering the specific glycosylation sites within proteins (Fig. 3f).

The low levels of HLA class I N-glycosylation can be explained by the fact that N-linked glycans are removed by N-glycanase before cytosolic proteins enter the cylindrical proteasome[41,42]. In contrast, N-glycans are known to withstand MHC class II antigen processing and remain attached to associated peptides, leading to alternative glycosylated products and truncated glycans in the immunopeptidome owing to lysosomal stress[41]. This phenomenon appears to reduce steric hindrance, providing an advantageous effect, and may explain the higher rate of glycopeptide truncation[41,43,44]. Overall, the data showed remarkable enrichment of glycosylation in HLA class II-associated peptides relative to HLA class I and the whole proteome, leading us to

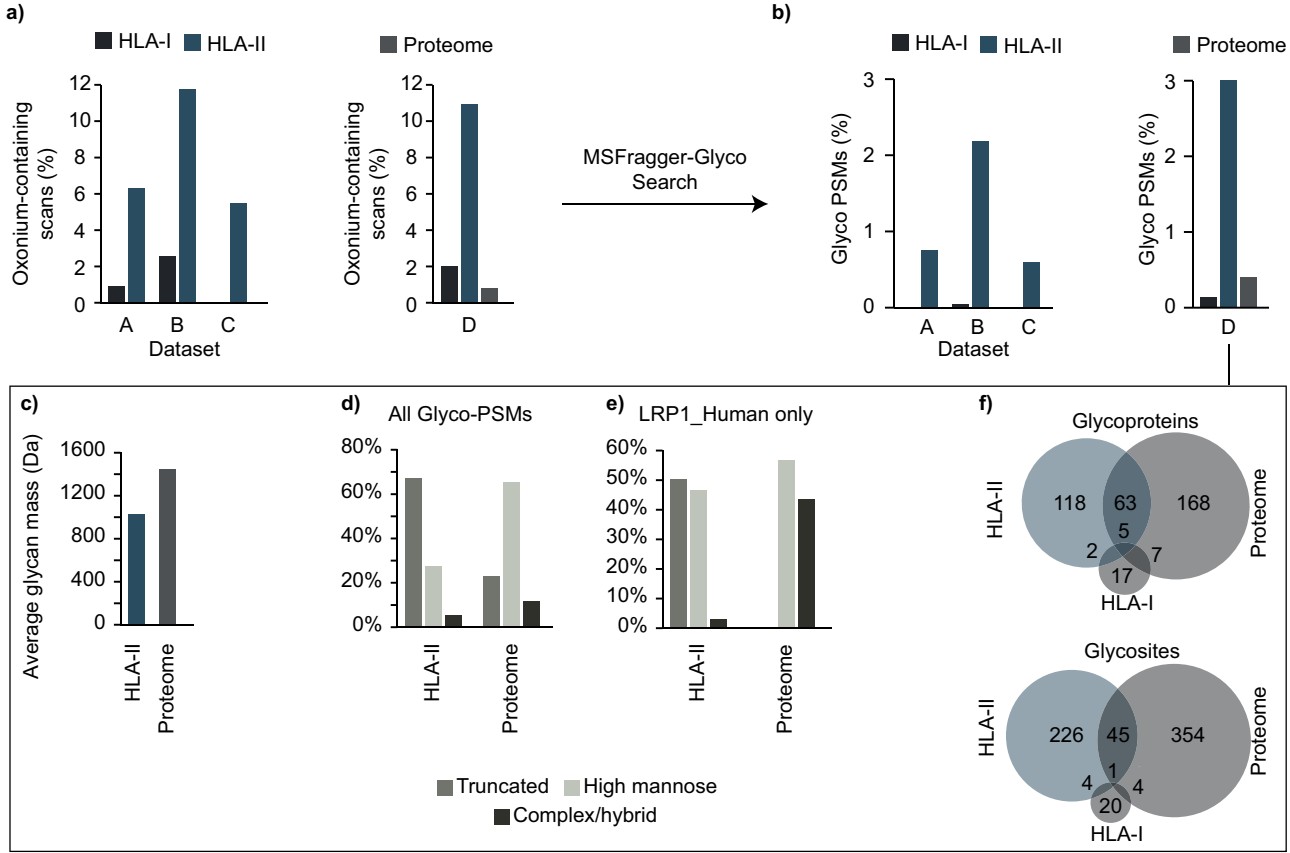

**Fig. 3 | Comparison of glycosylation at the proteome, HLA I, and HLA II peptidome levels. a** Levels of oxonium ions for HLA classes I and II in three datasets (A: Bassani-Sternberg et al.[31], B: Chong et al.[34], C: Marcu et al.[32]), along with the whole proteome in dataset D: Forlani et al.[37]. **b** Percentage of Glycosylated PSMs for the HLA class I and II immunopeptidome in 3 datasets (A, B, C), along with the whole proteome in dataset D. **c** Average glycan mass in Dalton (Da) for the HLA class II immunopeptidome versus the whole proteome in dataset D. **d** Glycan types for the class II immunopeptidome versus whole proteome in dataset D. **e** Glycan types found in the low-density lipoprotein receptor-related protein 1 (LRP1) for the class II immunopeptidome versus whole proteome in dataset D. **f** Comparison of glycoproteins (top) and glycosites (bottom) found in the HLA class I, II immunopeptidome, and whole proteome of dataset D.

focus the remainder of our efforts on HLA class II-associated and glycosylated peptides.

**Glycosylation of MAPs does not influence the HLA-binding motif**
To explore glycosylation in the context of HLA class II presentation, we focused on the HLA-binding core, a 9-mer sequence that interacts with the HLA molecule. In most mass spectrometry experiments, samples express multiple HLA alleles, leading to an ambiguous association between the identified peptides and the pool of available HLA molecules. Hence, a deconvolution step to find the HLA motifs and the corresponding binding core offsets of each peptide was deemed necessary for further experimentation (see "Methods").

We first chose to use MoDec[38] for deconvolution, a fully probabilistic framework that learns both motifs and the preferred binding core position offsets from the sequences themselves. The fact that MoDec does not rely on a pre-trained model is crucial when exploring HLA-bound peptides with post-translational modifications (i.e., glycosylation) to avoid the removal of peptides that are not well modeled. Such a deconvolution strategy requires manual intervention to choose the number of HLA motifs (i.e., the number of clusters) and assign each discovered motif to one of the HLA alleles expressed in a given sample. We carefully selected a case study of a human B-lymphoblastoid cell line (C1R) from Ramarathinam et al.[36]. The purification protocol of the HLA-bound peptides in this study was performed sequentially with pan anti-class I, class II anti-DP (Fig. 4a), class II anti-DQ (Fig. 4b), and class II anti-DR antibodies (Fig. 4c, d). Hence, the resulting mass spectrometry samples were mono-allelic (i.e., presenting one allele at a time), except

for the DR samples with the DRB1*12:01 and DRB3*02:02 alleles eluting together. Figure 4 presents 4 sections a, b, c, and d standing for the HLA class II alleles DPA1*02:01/02-DPB1*04:01, DQA1*05:05-DQB1*03:01, DRB1*12:01, and DRB3*02:02, respectively. All alleles showed a similar percentage of glycosylated and non-glycosylated peptides with the corresponding HLA motifs after deconvolution (Fig. 4, Panel I). It is worth noting that the lack of glycopeptides in the 5th DQ allele replicate is most likely due to the stochastic nature of mass spectrometry at the peptide level. Since the percentage of glycopeptides identified in unenriched data is generally low (<5%), the lack of identification in a replicate should not be taken as evidence of their complete absence. All 25 replicates showed an unaltered HLA-binding core with glycosylation (two-sided Fisher's exact test, 25 $P$ values > 0.05). Our results suggest that the presence of glycans on MAPs does not appear to significantly affect the binding interactions between HLA molecules and peptides, as evidenced by the high similarity in HLA-binding motifs between glycosylated and non-glycosylated peptides. However, it should be noted that this conclusion is limited to glycans that successfully bind to HLA molecules. Hence, further studies utilizing HLA glyco-enriched data and larger datasets are required to gain a more detailed understanding of the potential impact of glycans on HLA molecule-peptide interactions.

Considering the concordance of glycopeptide sequences with the HLA-binding cores, we checked the absolute glycosylation position per peptide length (i.e., glycosylation offset within the peptide). Figure 4 panel II shows a glycosylation tendency towards the N- and C-termini

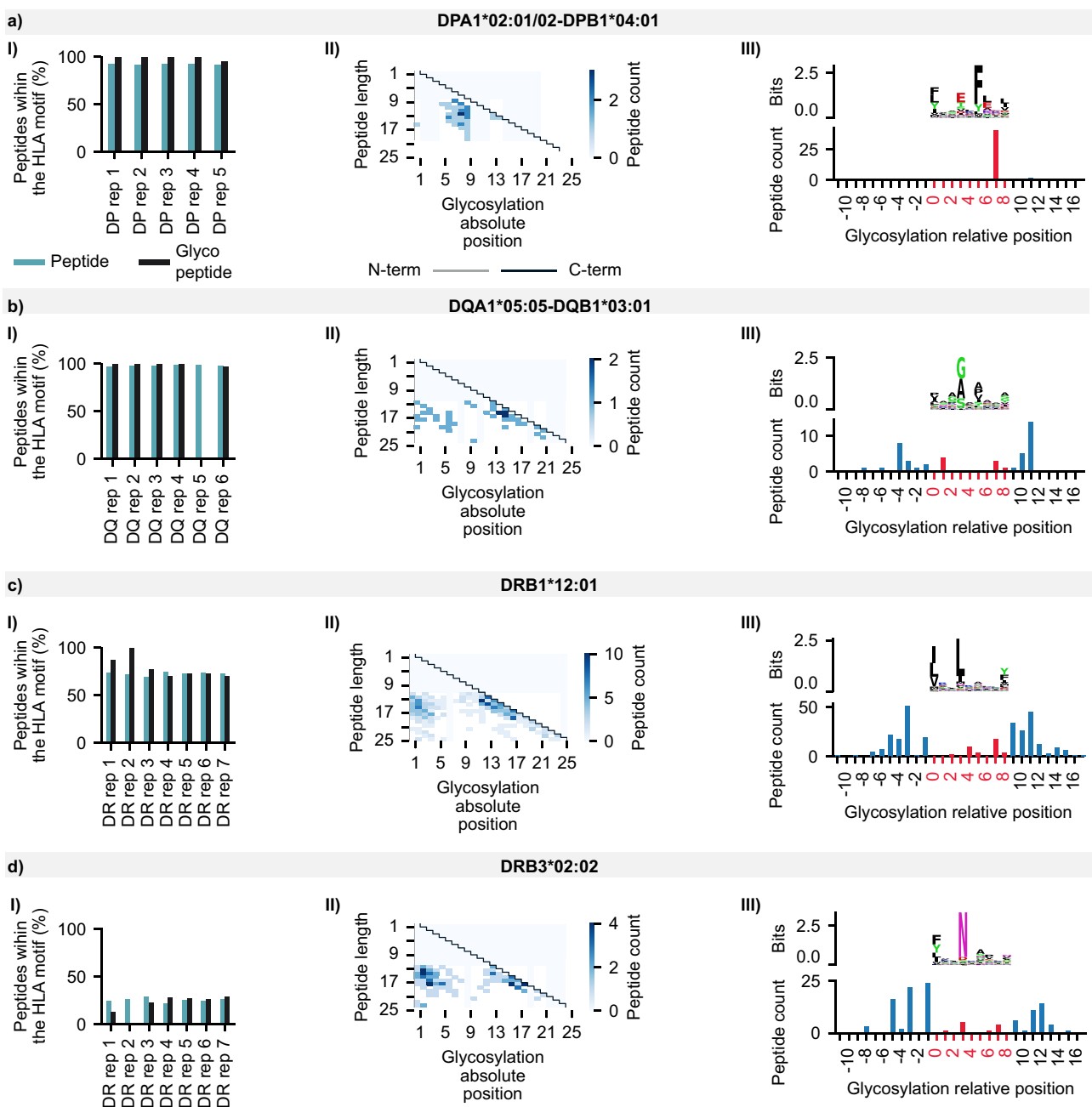

**Fig. 4 | Semi-supervised deconvolution of glycosylated HLA peptides from Ramarathinam et al.[36] using MoDec.** Panels I show the percentage of peptides and glycopeptides presenting the HLA-binding motif. Panels II display the glycosylation absolute position within the peptidic sequence (x-axis) and the peptide length (y-axis). Gray and black lines indicate the N-term and C-term, respectively, whereas the white to blue gradient represents the number of peptides with a specific glycosylation position at a specific peptide length. Panels III present the HLA-binding motif after deconvolution with MODEC (top) and the number of glycopeptides per relative glycosylation position (bottom). Negative values refer to glycosylation positions upstream of the HLA-binding core; values between 0 and 8 represent positions within the HLA-binding core; and values ≥ 9 refer to positions downstream of the HLA-binding core. Panel I includes both regular and glyco-search peptides, whereas Panels II and III show only glycopeptides from the union of all replicates per allele. **a** Peptides associated with the HLA allele DPA1*02:01/02-DPB1*04:01 of the C1R cell line. **b** Peptides associated with the HLA allele DQA1*05:05-DQB1*03:01 of the C1R cell line. **c** Peptides associated with the HLA allele DRB1*12:01 of the C1R cell line. **d** Peptides associated with the HLA allele DRB3*02:02 of the C1R cell line.

for both DQ and DR alleles (Fig. 4 sections b, c, and d in panel II) and only the C-terminal tendency for the DP allele (Fig. 4 section a in panel II). To further decipher glycosylation in the context of the HLA-binding cores, we examined the relative position shown in Fig. 4 panel III (i.e., glycosylation offset from the HLA-binding core start). Negative values indicate sites upstream of the HLA-binding motif start site, 0–8 indicate reference positions within the HLA-binding core, and values greater than 8 denote glycosylation sites downstream of the HLA-binding core. For the DPA1*02:01/02-DPB1*04:01 allele, glycosylation occurred 91% of the time within the HLA motif at position 8 (Fig. 4 section a, panel III). In contrast, for the other three alleles, glycosylation was more likely (86% of the time) to occur upstream or downstream of the HLA-binding core. These findings suggest that the position of glycosylation within the HLA-binding core is not constant but varies depending on the specific HLA allele. The DPA102:01/02-DPB104:01 allele is more likely to have glycosylation within the HLA-binding core, whereas the other alleles exhibit a higher likelihood of glycosylation upstream or downstream.

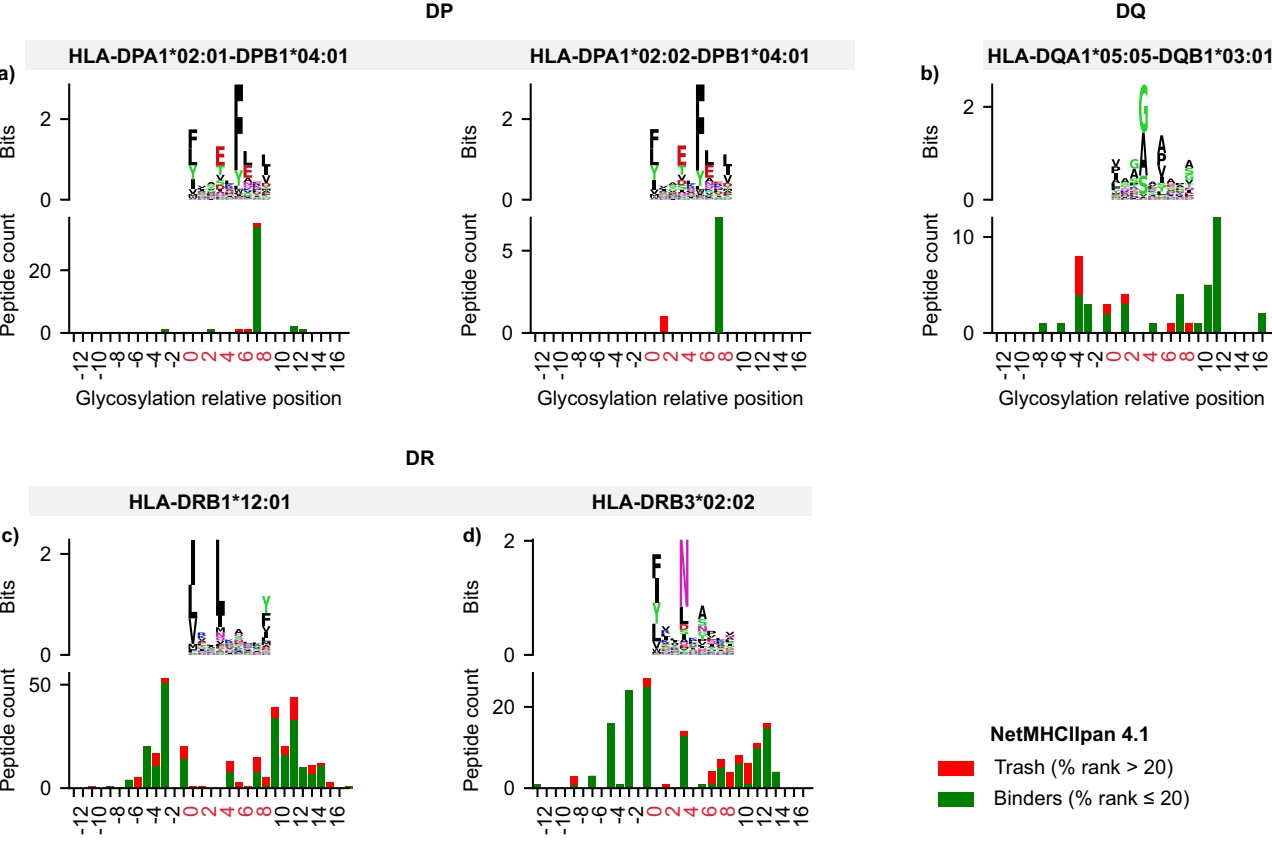

**Fig. 5 | Fully unsupervised deconvolution of glycosylated HLA peptides from Ramarathinam et al.[36] using NetMHCIIpan 4.1.** Each panel illustrates two levels of information: the top level shows the HLA-binding motif of peptides passing a NetMHCIIpan 4.1 percentile rank threshold of 20 after binding affinity prediction. The bottom level shows glycopeptides that are predicted to bind to a given allele in green (%rank ≤ 20); otherwise, non-binding peptides (i.e., trash) are shown in red (% rank > 20). Positions are shown relative to the HLA-binding core with negative values referring to the glycosylation position upstream of the HLA-binding core, values between 0 and 8 represent positions within the HLA-binding core, and values ≥ 9 refer to positions downstream of the HLA-binding core. **a** Deconvolution of glycosylated peptides associated with the HLA-DPA1*02:01/02-DPB1*04:01 alleles. **b** Deconvolution of glycosylated peptides associated with the HLA-DQA1*05:05-DQB1*03:01 alleles. **c** Deconvolution of glycosylated peptides associated with the HLA-DRB1*12:01 allele. **d** Deconvolution of glycosylated peptides associated with the HLA-DRB3*02:02 allele.

Despite the usefulness of MoDec for a previously unexplored category of peptides, such a tool suffers from several limitations:[45,46] (I) the need for manual intervention to associate the identified motifs with known allele specificities present in the sample; (II) the difficulty of assigning peptides to MHC molecules when alleles with overlapping motifs are co-expressed; (III) low sensitivity with low expression of MHC molecules; and (IV) the complexity of HLA class II specificities due to the involvement of the variable alpha and beta chains for the HLA-DQ and HLA-DP groups. These limitations render motif-allele assignment a daunting task, especially for up to 87 subjects in our dataset. Thus, we used the state-of-the-art binding model NetMHCIIpan 4.1[46,47] to perform MHC motif deconvolution and assign glyco-peptide sequences to their most likely HLA alleles without the need for manual intervention (see Methods). Consistently, glycosylated and non-glycosylated peptides from Ramarathinam et al.[36] showed similar binding properties, indicating that the detected glycosylation fits within the known HLA-binding cores (two-tailed Fisher's exact test, P value:0.48). Interestingly, NetMHCIIpan 4.1 confirmed most peptides with glycosylation were located at P8 within the HLA-binding core (97% for DPA1*0201 and 100% for DPA1*0202) for the C1R DP allele (Fig. 5a). Overall, 95%, 83%, 76%, and 87% of glycopeptides were found to bind to C1R DP (Fig. 5a), DQ (Fig. 5b), DRB1*12:01 (Fig. 5c), and DRB3*02:02 (Fig. 5d), respectively. Hence, we conducted the NetMH-CIIpan 4.1 deconvolution for the 83 remaining subjects in our dataset (see Supplementary Fig. 2 and Supplementary Data 4).

### The HLA class II N-glycosylation characteristics
We noticed a high tendency for glycosylation within the HLA-binding core for HLA-DP alleles, followed by a lower tendency for HLA-DQ, and even a lower tendency for HLA-DR alleles (see Supplementary Fig. 2). Hence, we checked for the occurrence of such events in each of the three HLA groups (DP, DQ, and DR). Figure 6a shows that up to 57% of HLA-DP-associated peptides had glycosylation inside the HLA-binding core, 30% for HLA DP, and 13% for HLA DR. In terms of glycan types, Fig. 6b shows that HLA DP-associated peptides had the highest fraction (0.67) of truncated glycans compared to DQ (0.55) and DR (0.41). High-mannose glycans showed a reverse trend for the DR, DQ, and DP alleles, with fractions of 0.37, 0.27, and 0.21, respectively. All DP-, DQ-, and DR-associated peptides showed a depletion in complex/hybrid glycans, in accordance with previous findings[9,19]. Unlike the DQ and DR alleles, glycosylation within the HLA-binding core appears to be associated with the DPA HLA gene. We speculate that this could be related to a DPA allele preference for binding truncated glycopeptides (smaller carbohydrate moiety), causing less hindrance in the pocket. However, we did not observe a significant difference in Dalton mass between the glycans located within and outside the HLA-binding core.

### Discussion
Post-translational modifications increase the diversity of the immuno-peptidome and may provide new targets for the immune system to recognize tumor cells or respond to pathogens. With PTM-driven

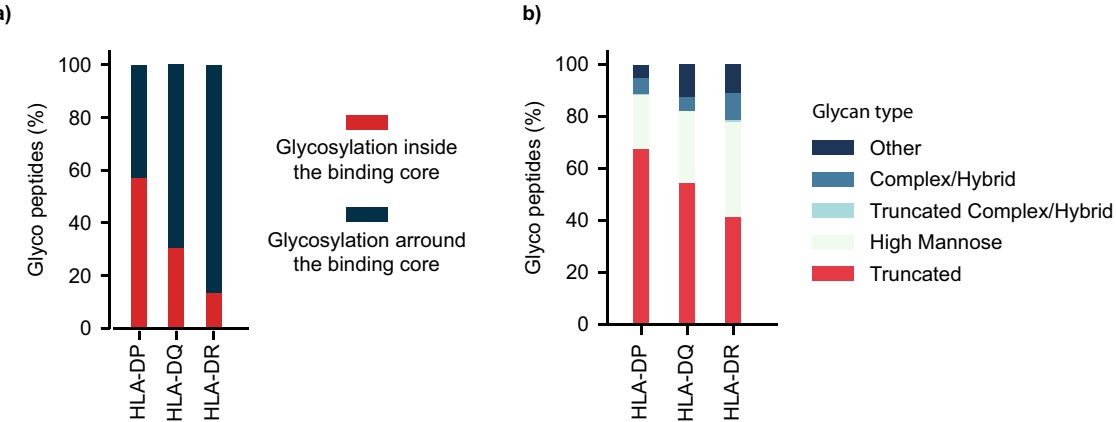

**Fig. 6 | Glycan characteristics of glycosylated HLA class II-associated peptides. a** Percentage of glycosylation inside (red) and outside (blue) the HLA-binding motif per HLA group (DP, DQ, and DR). **b** Distribution of glycan types among the studied HLA class II groups (DP, DQ, and DR).

antigenicity being continuously highlighted[9,31,48,49], glycosylation is a key PTM that, despite its long history of research, remains understudied in the context of MHC presentation due to computational challenges.

This study presents a methodology to analyze HLA N-glycopeptides using MSFragger-Glyco. Specifically, we have developed a glycan-group FDR method and fine-tuned a glyco-immunopeptomics analytical workflow. These improvements enabled effective interrogation of nonenzymatic and non-enriched glycosylation data at a large-scale. We used this workflow to produce a resource of HLA class II N-glycosylated MAPs arising from a harmonized analysis of eight publicly available studies. Overall, we identified 1049 glycosylation sites from 3409 different glycopeptides, an order of magnitude greater than any previous effort in this area. Leveraging this large-scale resource, we explored the properties of glycosylated MAPs, including the types of glycans conjugated, MHC binding affinity predictions, and the positioning of glycosylation relative to the HLA-binding core.

Previous research by Malaker et al.[9] covered five DR alleles and showed that only 3 out of 23 peptides had glycosylated residues within the binding core. To better understand the implications of these findings, the authors used molecular modeling to postulate that glycan residues are most likely to protrude from the HLA-binding pocket and interact with the complementary determinant region of the T-cell receptor. Our study provides a comprehensive analysis of HLA class II N-glycosylation across a larger set of HLA alleles than previously examined. We expand the coverage to 28 DR alleles and includes multiple DP and DQ alleles, resulting in a total of 87 HLA molecules when both alpha and beta chains are considered. In addition to confirming the previous finding of a preference for terminal glycosylation of peptides associated with DR and DQ alleles, our study reveals an HLA-binding core glycosylation tendency for peptides associated with DP alleles. These findings suggest that the involvement of glycan residues in the HLA-binding pocket is more complex and varied than previously believed. While our current evidence suggests that glycosylation does not affect HLA-binding cores, a more complete understanding of the potential impact of glycopeptides on HLA molecule-peptide interactions would benefit from the availability of HLA glyco-enriched data, an experimental technique that remains challenging and has yet to be accomplished. Moreover, larger datasets could facilitate the identification of altered HLA-binding cores above the level of background noise. Therefore, it is possible that our analysis did not account for peptides harboring glycans that could modify peptide interactions with HLA molecules, highlighting the need for further research in this area.

It has been suggested that N-linked glycans must be removed by N-glycanase before cytosolic proteins enter the cylindrical-shaped proteasome[41,42]. This may explain our observation of the lack of HLA class I N-glycosylation, considering that the major source of MHC class I

proteins is the cytosol. In contrast, antigen-presenting cells take up glycoproteins via endocytosis and transport them to lysosomal compartments, where proteolytic enzymes with low pH fragment them into peptides. These peptides bind to empty MHC class II molecules, forming stable MHC-peptide complexes. Studies have shown that N-glycans withstand this process and remain attached to class II MAPs[41,43,44].

Glycosylation plays a crucial role in antigen processing, as it can affect proteolysis by obstructing proteases through steric hindrance. Our study revealed variations in glycan types across HLA groups (DP, DR, and DQ), with DP alleles exhibiting an enriched presence of truncated glycans, and DR alleles demonstrating a higher mannose content. We also found that DP alleles have a higher proportion of glycans within the binding core (57%), compared to DQ alleles (30%) and DR alleles (13%). MHC class II processing can alter the glycan segments and result in alternative glycosylated products[41]. Glycan truncation may provide an advantageous effect by reducing steric hindrance, leading to the higher occurrence of truncated glycans in the class II immunopeptidome than in the proteome. This may also explain the higher rate of glycopeptide truncation associated with DP alleles, as 57% of these were located within the HLA-binding core. Although glycans enhance the likelihood of presentation by making peptides more resistant to proteolysis, larger and complex glycan structures are not well recognized by T cells, as observed in previous research[41]. It is worth noting that this analysis may not have accounted for the immune evasion that glycosylation could provide, highlighting the need for further exploration. Moreover, future research should investigate whether the correlation between smaller glycans and their presence within the HLA-binding core is solely due to size restrictions or whether it reflects other processing of MAPs for presentation.

Enrichment of glycosylated peptides on MHC-II while preserving canonical binding motifs offers the possibility of designing and developing glycosylated neoantigen vaccines with improved affinity over wild-type peptides[25,26]. This is further notable given that most of the known antitumor CD4+ T cells are specific for highly immunogenic self-derived MHC-II antigens, demonstrating that self-antigen CD4+ T cells can mount antitumor responses. Cancer-specific glycosylation of MAPs may further contribute to the restriction of these mechanisms to the tumor microenvironment. We have made our findings readily available as a web resource to query pertinent information regarding the identified glycosylated MAPs. Users can search for a specific glycan, MAP sequence, protein, or glycosylation site that is associated with a specific HLA allele. In addition, we included deconvolution information, allowing further interpretation of the data within the HLA haplotype context. We are planning to develop this initiative, introduce more studies, and increase the HLA allele coverage. Moreover, by providing an optimized computational workflow file that can be loaded directly into FragPipe to

reproduce the method described here, we make it easy for others to carry out challenging glyco-imunopeptidomics analyses on new datasets. Our pipeline is broadly accessible and can be executed on typical local workstation computers provided sufficient RAM is available (a minimum of 64 GB of RAM is required). The analysis of the eight large HLA class II datasets took 52.3 hours, or 2.7 min per raw file, to complete on a server equipped with 28 cores and 512 GB of RAM. Of the total processing time, 80% was devoted to MSFragger search, 9% to validation and FDR filtering, and 11% for quantitation. These results demonstrate the feasibility of our approach for large-scale, comprehensive analysis of glycosylation patterns in non-enriched datasets, which is critical for understanding the glycoproteome in biological samples.

With our aim to support tumor antigen discovery and broaden potential targets, we developed a workflow to study N-glycosylated MAPs. Moreover, we have recently added an MSFragger-Glyco-compatible version of O-Pair[50,51] to FragPipe, enabling confident annotation of O-glycosylated MAPs. However, the scarcity of electron-based activation datasets required to locate O-glycans currently limits its large-scale application. In summary, we anticipate that the findings of this work will spark increasing interest in glycosylation studies within the field of immunopeptidomics.

## Methods

### Dataset selection
Studies from the PRIDE[52] database were first screened based on a list of keywords related to immunopeptidomics (Supplementary Note 1). Low-resolution analyses were eliminated, and MHC-related datasets conducted with at least one of the following instruments were kept: Orbitrap fusion lumos/fusion, Q Exactive, LTQ Orbitrap, Orbitrap Exploris 480, TripleTOF, impact II, and maXis. Subsequently, manual curation of the resulting 312 studies was performed to filter non-relevant datasets, resulting in 140 HLA Class I, II, or I and II datasets. The number of proteins identified per study was retrieved from gpmDB[53], and datasets with a high number of protein groups were prioritized. A final manual curation step resulted in the selection of eight datasets included in this study (Supplementary Data 2).

### Mass spectrometry N-glycan search
Raw and wiff files were first downloaded from PRIDE and converted to the mzML format using msconvert[54] v3.0.20287 with peak picking, FragPipe (TPP) compatibility, and removal of zero-value filters. The analysis was executed within the FragPipe suite v18.1-build5 using headless mode. Glyco searches were performed using MSFragger v3.5 with methionine oxidation, N-terminal acetylation, and cysteinylation as variable modifications, and a list of 198 glycans (Supplementary Data 5). A list of contaminants was added to the UniProt Swiss-Prot (UP000005640) proteins[55] along with their corresponding reversed decoy sequences. Enzymatic cleavage was set to non-specific with peptide lengths from 7 to 25 amino acids for the eight HLA class II datasets, from 7 to 12 amino acids for HLA class I datasets (A, B, C, D), and fully enzymatic cleavage with peptide lengths from 7 to 50 amino acids for the whole proteome dataset D. Peptides containing the consensus sequon (N-X-S/T) and decoy (reversed) peptides containing the reversed sequon were considered as potential glycopeptides to ensure that the same number of potential glycopeptides was searched in both target and decoy databases. We confirmed that score distributions for decoys and false targets for peptides with and without sequons are highly similar (Supplementary Fig. 3). Only spectra containing oxonium ion peaks with a summed intensity of at least 10% of the base peak were considered for glycan searches, whereas all others were searched without considering glycosylation. Data were deisotoped[56] and decharged in MSFragger-Glyco, calibrated, and searched with 20 ppm mass tolerances for precursors and 15 ppm for products with MSFragger's built-in parameter optimization performed for each study[57]. Errors in monoisotopic peak detection by the instrument were allowed (+1 Da and +2 Da).

### FDR control
Filtering was performed using Philosopher[58] (v4.5.1-RC10), including PeptideProphet[59] modeling of peptide probabilities, ProteinProphet[60] protein inference, and Philosopher's internal filter for FDR control. The semi-parametric modeling of PeptideProphet was used, with the expectation value as the only contributor to the f-value. The number of tolerable termini (ntt) model was disabled, as it is not applicable to nonenzymatic searches. Filtering was performed in Philosopher using a modified group-specific FDR procedure. Non-glycosylated and glycosylated PSMs were filtered separately using a delta mass cutoff of 145 Da (the size of the smallest glycan considered in the search) to distinguish glycosylated PSMs from non-glycosylated PSMs. This allowed different score thresholds to be used to filter glycosylated and non-glycosylated PSMs to a 1% FDR. This is essential as the large search space for glycosylated PSMs results in higher scoring false matches, requiring a higher score threshold for effective filtering than for non-glycosylated PSMs. Since non-glycosylated PSMs constituted the majority of the results, filtering all PSMs together would yield an insufficiently low score threshold for glycosylated PSMs. After the group-specific 1% FDR filter was applied to the glycosylated and non-glycosylated PSMs, 1% peptide- and protein-level FDR filters were applied. A sequential filtering step was then applied to remove any PSMs matched to proteins that did not pass the 1% protein-level FDR. Glycan assignment was subsequently performed in PTM-Shepherd using the default N-glycan database[30] and parameters along with a 0.05 glycan q-value threshold.

### Deconvolution of the MHC-associated peptides
Motif deconvolution is the process of identifying HLA-binding motifs and their corresponding binding core offsets for a set of peptides. A first deconvolution that required manual inspection was performed using MoDec[38]. The peptides were grouped by subject (i.e., instances of the same replicates; see Supplementary Data 2, samples' sheet). A maximum of 10 clusters, 20 runs, and minimum peptide length of 12 amino acids were considered. Since HLA-II ligands from the same subject come from different alleles, MoDec provides a direct interpretation and assigns peptides with similar binding cores to clusters (i.e., HLA motifs). However, manual inspection is still required to (I) set the number of HLA motifs that MoDec detects per subject, and (II) annotate these motifs (i.e., clusters) to their respective HLA II alleles. Hence, MoDec-identified HLA motifs were assigned to the correct HLA class II alleles by manual inspection for each analyzed subject. A second deconvolution, inspired from Kaabinejadian et al.[46], which did not require manual inspection, was performed using NetMHCIIpan 4.1[47]. Briefly, all unique peptides were predicted for MHC presentation to all MHC alleles expressed in each subject. The likelihood of peptides being presented by a given MHC molecule is determined by the percentile rank score, which ranges from 0 to 100, with 0 being the strongest binding score. Peptides with a percentile rank score > 20 were considered MS co-immunoprecipitated contaminants and labeled as trash. Peptides with a percentile rank score ≤ 20 were assigned to the lowest scoring allele of a given subject. We applied the second deconvolution method using NetMHCIIpan 4.1 to the entirety of the subjects in this study, considering the similarity of the results to the first deconvolution method (i.e., MoDec).

### Data processing and figure generation
The analysis was performed in an Anaconda Python environment with the following packages: Python (version 3.9), Biopython (version 1.79), Matplotlib (version 3.5.1), Plotly (version 4.14.3), Scikit-learn (version 1.0.2), Pandas (version 1.4.1), Pyteomics (version 4.5.3), Matplotlib-venn (version 0.11.6), Seaborn (version 0.11.2), UMAP-learn (version 0.5.2), HDBSCAN (version 0.8.28), and xlrd (version 2.0.1), Logomaker v0.8. An R environment within Anaconda was also used, consisting of R-essentials (version 4.1), R-base (version 4.1.2), and R-devtools (version 2.4.3). These tools and environments enabled data processing and creation of visualizations.

## Reporting summary

Further information on research design is available in the Nature Portfolio Reporting Summary linked to this article.

## Data availability

The human reviewed protein sequences used in this study are available in the UniProt database under accession code UP000005640. The *Arabidopsis thaliana* reviewed protein sequences used in this study are available in the UniProt database under accession code UP000006548. The proteomic glycosylation data used in this study are available in the GlyGen database (https://www.glygen.org).The mass spectrometry data used in this study are available in the PRIDE database under accession codes PXD006939, PXD020079, PXD014017, PXD019643, PXD020186, PXD025877, PXD004894, PXD013649, PXD012308 and PXD005565 (yeast enriched N-glycopeptide data). The processed immunopeptidomics data is available at https://hla-glyco.nesvilab.org. Source data are provided with this paper.

## Code availability

The code used to reproduce the findings of this study is available on the following GitHub repository: https://github.com/Nesvilab/HLA_glyco_analysis.

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

## Acknowledgements

This work was funded by NIH grants R01-GM-094231, U24-CA210967, U24-CA271037 received by G.B., D.P., Y.H., F.Y., F.L., M.C., A.N. This work was supported by the International Centre for Cancer Vaccine Science (Fundacja na rzecz Nauki Polskiej: MAB/3/2017) project is carried out within the International Research Agendas program of the Foundation for Polish Science co-financed by the European Union under the European Regional Development Fund (received by G.B. and J.A.). This project has received funding from the European Union's Horizon 2020 research and innovation program under grant agreement No 101017453 (received by J.A.).

## Author contributions

G.B. collected and curated the data, generated the figures, supplementary materials, and drafted, coordinated, and revised the manuscript. D.P. performed the immunopeptidomic analysis, supported figure generation, interpreted the results, and contributed to the drafting, revision, and coordination of the manuscript. Y.H. produced the web portal and assisted in revising the manuscript. F.Y. supported the study with software development tasks. F.L. supported the study by adding a group-specific FDR feature to Philosopher. J.A. contributed to the study by revising the manuscript. M.C. helped with the study design and manuscript revision. A.N. conceived the project, assisted with the study design and manuscript revision, and provided overall supervision.

## Competing interests

The authors declare no competing interests.
