## [Peer Review File · Nature Communications]

Unraveling the glycosylated immunopeptidome with HLA-GlycoReviewers' Comments:

Reviewer #1:

Remarks to the Author:

The manuscript by Bedran and colleagues describes a new tool to identify MHC-bound glycopeptides based on the MSfragger architecture. This was a well written and excellent paper and I have no criticism of the approach and the underlying informatics.

My only critique relates to the results section that although predominantly descriptive should be qualified – comparison of different datasets that use different instruments and acquisition methods must be taken into account when generalities are made concerning the nature of HLA-I and HLA-II bound glycopeptides and across healthy and diseased states. The authors must comment on this – in particular the acquisition parameters for proteomics, HLA-I and HLA-II are often optimized with larger or smaller precursor ions excluded from MS/MS analysis. This may in turn influence the nature of glycopeptides sampled by MS/MS. Moreover, fractionation of the sample off-line often produces deeper coverage and therefore also samples more low abundance species which may include some glycol-immunopeptides. I would therefore encourage the authors to qualify their statements with this in mind.

Finally I congratulate the authors on producing another publically accessible tool and for their support of the growing immunopeptidomics community.

Reviewer #2:

Remarks to the Author:

Tsai et al. presented a computational glycol-immunopeptidomics workflow that exploits fast processing time of MSFragger and clustered FDR estimation method to maximize glycopeptide ID sensitivity. By using pre-filtration of both input data - proteome database (retaining the peptides containing N-X-S/T sequon) and spectra (the ones containing oxonium ions) - and by optimizing the post processor (PeptideProphet) and the FDR estimation method, an order of magnitude larger number of glycopeptides are identified than the previously reported ones. With the introduced workflow, the authors focused on the analysis of glycosylated MAPs and also generated HLA-Glyco, a resource containing thousands of HLA class II N-glycopeptides. The analysis reveals several biologically implacable insights on HLA glycopeptides. While the manuscript is written in a concise way in general, a few points need to be addressed to be published in Nature comm. journal. In particular, the results section corresponding to Fig 4 - 6 could be supplemented with more interpretation on the observations.

Major points

1. In introduction, the authors stated "building on these advances, we developed an optimized workflow for HLA glyco searches with a focus on optimizing the false discovery rate (FDR) control of glycosylated MAPs." But additional description on FDR control for glyco searches should be added in front of this sentence to make the text more logical. What kind of FDR control other than plane target decoy has been tested for glyco searches and why it failed should be described. I believe the authors gave similar explanation after this sentence.

2. In glycan searches, the authors used peptides containing the consensus sequon (N-X-S/T) and decoy (reversed) peptides containing the "reversed sequon" to ensure the that same number of potential glycopeptides was searched in both target and decoy databases. I believe this could possibly cause bias on FDR estimation. The decoy peptides should represent false positive glyco peptides that containing the normal sequon. But all decoys contain the reversed sequon. And by the filtration scheme that the authors employed, the spectra are mostly from the glyco peptides and thus likely to contain the peaks matching with normal sequon sequences. Thus the decoy with the reversed sequon

sequences is likely to have lower match score than the false positive glyco peptides. It may be too much to generate all the results from the scratch at this stage, but the observation that the decoys with normal sequon sequences (which could be generated by reversing other amino acids than the amino acids corresponding to the sequon sequences, for instance) and the currently used decoys generate comparable number of IDs have to be provided, for instance in a supplement. And I would suggest to use this new decoy should be the standard in this workflow in the future.

3. Overall, for readers (including me) who are not very familiar with immunopeptidomics, the text corresponding to Fig 4-6 would read like listing of observations. While deeper interpretation is given in discussion section, short but enlightening explanation on the each result could be very helpful for readers to understand the text and thus to attract more interests and in turn citations. The readers of Nature comm. are not limited to the mass spec experts nor to immunopeptidomics experts.

4. This workflow identified more than 1,000 new glyco sites, which is orders of magnitude higher number of IDs. But general analysis on this newly identified sites is missing. The authors focused on analysis of glycopeptides in the context of HLA presentation. A general analysis on these 1,000 IDs demonstrating that most of them are not all false positives should be provided, at least.

Minor points

1. In Fig 1, the regular search results are in grey color. Does this imply this result is not used in the whole analysis? As far as I understood, they are used in the following analyses, like Fig 4. Panel i. In the legend or in text, please specify how the results are used briefly.

2. Related to Fig3. F, a quantitative analysis (like in Fig.2 F) on HLA-II and proteome exclusive glycoproteins should be provided to justify "The whole proteome glyco search likely captures glycopeptides from the most abundant glycoproteins, as the experiment was performed without any glycopeptide enrichment, whereas the immunopeptide datasets presumably capture MAPs with much less dependence on overall protein abundance. "

3. In Fig 4 Panel I, the peptides and glyco peptides are from the normal and glyco searches in Fig 1? And Fig4. ii and iii are only for the Glyco peptides in Fig 4. i? Even if it may be obvious, it is always helpful to clarify the scope of the data that are used to generate a specific subfigure.

4. Likewise, Fig 4 ii are the peptides the ones jointly found in replicates in Fig 4 i? Or the union? From Fig 4 i, it seems that each allele has many replicates. But again the scope of the data used to generate figures is not clear.

5. In Fig4. iii, for the first DPA allele, glycosylation and HLA binding seem to have a strong connection while for others not. Could there be any relevant interpretation for this localizations? The results section corresponding to this figure just ends with the observation and thus it is hard to get the take home message from this part.

6. Fig. 4b panel I. DQ rep5 Glycol peptide percentage is shown to be zero. Is this an error in figure? And what is the interpretation for the similar percentages? In the discussion section, the authors mentioned "interestingly, we observed no difference in binding motif predictions with glycopeptides compared to non-glycopeptides, despite some peptides containing glycans within the binding core." Is it interesting because this observation is against a well known prediction? If so, could there be relevant interpretation on this? The discussion section could be supplemented with more data interpretation as well.

7. Fig. 5 and 6 also lack biological interpretation.

Reviewer #3:

Remarks to the Author:

This manuscript performed glycosylation analyses of MAPs from 8 publicly available studies and created HLA-Glyco, a resource of HLA N-glycopeptides. Identifying glycosylated MAPs is surely important for the exploration of additional source of specificities targeted by immune responses to tumors or pathogens. While I see a potential resource aspect of the paper, unfortunately I do not think that the technical advances and findings presented have a sufficiently significant. Everything

they seem to be doing is analyzing the public data with MSFragger-Glyco.

The authors stated that they developed a novel computational glyco-immunopeptidomics workflow and used one figure (Figure 1) to describe the workflow, but they indicate nothing specific to this: in what way is this more advanced than MSFragger-Glyco? It seems that the research is primarily the use of MSFragger-Glyco. Optimizing the FDR control may be not fully representative of progress, as glycopeptide FDR control has already been reported by MSFragger-Glyco (Mol. Cell. Proteomics 2022, 21, 100205), and also by several other software tools, such as GP Finder (although only for simple samples), pGlyco3, Glyco-Decipher, et al. I think they should clearly point out that if it is just an application research or there is the novelty of their glyco-immunopeptidomics workflow, as well as the improvements and differences from previous software. Benchmark with similar software would be convincing.

The authors used the public MS data conducted without any glycopeptide enrichment. I think the identification will largely depend on the overall abundance, and the large amount of nonglycopeptides will interfere the depth and sensitivity of glycopeptide identification. Why not specially perform experiments to enrich HLA glycopeptides?

Interpretation of glycopeptides from proteome or immunopeptide datasets requires high search sensitivity, speed and precision. The authors analyzed 2,000 LC-MS/MS runs and found 3409 class II N-glycosylated MAPs. I am curious about the search time and space. In addition, the depth and comprehensiveness of the identification will greatly influence the subsequent analyses and conclusions. Thus, to fully mine these public data, the comprehensive comparisons of the identifies from different MS data and widely-used software tools are highly suggested.

They performed insufficient depth of the result mining and analysis, from which, we could get very limited information. For example, one of the main findings is that they find a higher percentage of truncated glycans (68%) in the HLA-II sample. While no other exploration was made. The quantitative and heterogeneity analysis of these glycans are necessary for us to characterize these HLA N-glycopeptides. For HLA-glyco data, the authors mainly focus on the N-glycosylation-driven HLA motif analysis. However, glycan-related motif information was ignored. Besides, conclusions, such as sample specific signatures, unique glyco-antigens, are also hard to get from the current analyses.

The O-glycosylated MAPs are also of potentials interest as described in the paper. In the discussion part, the authors should objectively discuss the difficulties and the key points of O-glycosylated MAPs identification rather than simply saying that "in principle, be studied by our method", since there is no data to support this conclusion. I do not think O-glycosylated MAPs can be easily interpreted from these public MS data using the workflow presented here.

Response to reviewers' comments on the manuscript "HLA-Glyco: A large-scale interrogation of the glycosylated immunopeptidome" (Reference number NCOMMS-22-45486-T).

Reviewer 1

Comment 1: The manuscript by Bedran and colleagues describes a new tool to identify MHC-bound glycopeptides based on the MSfragger architecture. This was a well written and excellent paper and I have no criticism of the approach and the underlying informatics.

We thank the reviewer for the positive comment.

No action necessary

Comment 2: My only critique relates to the results section that although predominantly descriptive should be qualified – comparison of different datasets that use different instruments and acquisition methods must be taken into account when generalities are made concerning the nature of HLA-I and HLA-II bound glycopeptides and across healthy and diseased states. The authors must comment on this – in particular the acquisition parameters for proteomics, HLA-I and HLA-II are often optimized with larger or smaller precursor ions excluded from MS/MS analysis. This may in turn influence the nature of glycopeptides sampled by MS/MS. Moreover, fractionation of the sample off-line often produces deeper coverage and therefore also samples more low abundance species which may include some glycol-immunopeptides. I would therefore encourage the authors to qualify their statements with this in mind.

Response: We appreciate the reviewer's constructive critique and attention to the intricacies of using different datasets, with varying instruments and acquisition methods. We agree that narrow precursor selection windows often used in immunopeptidomics acquisition methods could bias the glycopeptides/glycans observed and have revised the corresponding statements to make this clear.

We added the sentence below to the manuscript.

The absence of glycosylated HLA class I MAPs in the healthy samples of dataset C is intriguing. Nonetheless, it is worth considering that the data was obtained using a very restricted precursor m/z range, which could have led to the exclusion of glycopeptides from the analysis due to their higher molecular weight.

The new changes can be found at lines 195 - 198 in the revised manuscript.

Comment 3: Finally I congratulate the authors on producing another publically accessible tool and for their support of the growing immunopeptidomics community.

We thank the reviewer for the positive comment.

No action necessary

Reviewer 2

Comment 4: Tsai et al. presented a computational glyco-immunopeptidomics workflow that exploits fast processing time of MSFragger and clustered FDR estimation method to maximize glycopeptide ID sensitivity. By using pre-filtration of both input data - proteome database (retaining the peptides containing N-X-S/T sequon) and spectra (the ones containing oxonium ions) - and by optimizing the post processor (PeptideProphet) and the FDR estimation method, an order of magnitude larger number of glycopeptides are identified than the previously reported ones. With the introduced workflow, the authors focused on the analysis of glycosylated MAPs and also generated HLA-Glyco, a resource containing thousands of HLA class II N-glycopeptides. The analysis reveals several biologically implacable insights on HLA glycopeptides. While the manuscript is written in a concise way in general, a few points need to be addressed to be published in Nature comm. journal. In particular, the results section corresponding to Fig 4 - 6 could be supplemented with more interpretation on the observations.

Response: We thank the reviewer for their positive feedback on our computational glyco-immunopeptidomics workflow and for pointing out the need for additional interpretation in the Results section, corresponding to Figures 4-6. We are glad that they have found our manuscript concise and that our biological results are insightful. We have revised the Results section to provide a more comprehensive interpretation of the observations, including additional insights into the significance of the results.

All specific revisions addressing this concern are listed below.

Comment 5 (major point): In introduction, the authors stated “building on these advances, we developed an optimized workflow for HLA glyco searches with a focus on optimizing the false discovery rate (FDR) control of glycosylated MAPs.” But additional description on FDR control for glyco searches should be added in front of this sentence to make the text more logical. What kind of FDR control other than plane target decoy has been tested for glyco searches and why it failed should be described. I believe the authors gave similar explanation after this sentence.

Response: We appreciate the reviewer’s concern regarding the clarity and logic of our text, and we agree that an additional description of FDR control for glyco searches would be beneficial to the reader at the Introduction level. We have clarified the critical distinction between peptide and glycan-level FDR analyses and our approaches to each one in this study. We have provided more information on FDR control for glyco searches, including a description of the other FDR control methods that have been tested and why they failed, as depicted in the two paragraphs below.

The large-scale analysis of glycosylated MHC-associated peptides (MAPs) presents significant challenges, primarily due to the enormous search space of glycosylated non-enzymatic peptides. To address these challenges, our recent developments in improving search speed²⁰ (MSFragger) and addressing the complexity of glycosylation²¹ (MSFragger-Glyco) have proven to be beneficial. However, it is worth noting that currently, glycosylated

MAPs are queried from data not enriched for glycosylated peptides, which poses a challenge as available false discovery rate (FDR) strategies are not suitable for this type of data. In light of this, we leveraged these advances to optimize a workflow for HLA glyco searches and developed an improved FDR strategy specifically for glycosylated MAPs obtained from non-enriched MS-based immunopeptidome data.

We evaluated several FDR strategies, and implemented a group-specific glycopeptide FDR approach that filters glycopeptides and regular peptides separately, allowing for different cut-off scores in each group. To further refine the results and ensure the accuracy of the identified glycans, we applied additional glycan-level FDR filtering (glycan q-value threshold of 0.05) after peptide-level filtering to remove glycoPSMs without strong spectral evidence for the glycan.

The new changes can be found at lines 57 - 73 in the revised manuscript.

Comment 6 (major point): In glycan searches, the authors used peptides containing the consensus sequon (N-X-S/T) and decoy (reversed) peptides containing the “reversed sequon” to ensure that the same number of potential glycopeptides was searched in both target and decoy databases. I believe this could possibly cause bias on FDR estimation. The decoy peptides should represent false positive glyco peptides that containing the normal sequon. But all decoys contain the reversed sequon. And by the filtration scheme that the authors employed, the spectra are mostly from the glyco peptides and thus likely to contain the peaks matching with normal sequon sequences. Thus the decoy with the reversed sequon sequences is likely to have lower match score than the false positive glyco peptides. It may be too much to generate all the results from the scratch at this stage, but the observation that the decoys with normal sequon sequences (which could be generated by reversing other amino acids than the amino acids corresponding to the sequon sequences, for instance) and the currently used decoys generate comparable number of IDs have to be provided, for instance in a supplement. And I would suggest to use this new decoy should be the standard in this workflow in the future.

Response: We appreciate the reviewer’s concern regarding the use of a reversed sequon when selecting decoy peptides. We would like to clarify in this response why our approach correctly estimates the FDR. Target-decoy approaches fundamentally depend on the assumption that both the target and decoy peptides have an equal chance of being matched to a spectrum by chance. Here, we filter the possible targets (forward sequences) using the forward sequon and possible decoys (reversed sequences) using the reversed sequon before matching the spectra. Our approach ensures equal proportions of glycopeptides in both the target and decoy space. Otherwise, the target-decoy assumption would be invalid, as target glycopeptides would be more likely to match a spectrum by chance than decoys, resulting in incorrect FDR estimation. Our approach of using decoy peptides containing the reversed sequon was carefully considered and has been shown to provide accurate and reliable results in previous studies (please check the MSFragger-Glyco paper for the FDR/reliability: <https://doi.org/10.1038/s41592-020-0967-9>).

We added the paragraph below to the Results section for clarity:

Glycopeptides typically comprise a sizable portion of the acquired spectra in the context of

glycopeptide enrichment, allowing the application of a single FDR strategy for all spectra. The previously published MSFragger Glyco workflow, which is tailored for large-scale glycoproteomics experiments, applies a target-decoy approach in which reversed N-glycan sequons are verified in reversed (decoy) peptides to guarantee an equal proportion of possible glycopeptides in the target and decoy search spaces. We have demonstrated (Polasky et al. 2020) that this approach can accurately and efficiently identify glycopeptides from the enriched data. Despite the significant benefits of glycopeptide enrichment, it is not yet feasible for MHC-associated peptide analysis owing to technical challenges that have yet to be overcome. This is mostly due to the large amounts of starting material required for the isolation protocol in immunopeptidomics experiments. In this case, the identification of glycopeptides becomes even more challenging, particularly in non-enzymatic searches, as glycopeptides represent a minority of acquired spectra.

The new changes can be found at lines 105 - 117 in the revised manuscript.

Comment 7 (major point): Overall, for readers (including me) who are not very familiar with immunopeptidomics, the text corresponding to Fig 4-6 would read like listing of observations. While deeper interpretation is given in discussion section, short but enlightening explanation on the each result could be very helpful for readers to understand the text and thus to attract more interests and in turn citations. The readers of Nature comm. are not limited to the mass spec experts nor to immunopeptidomics experts.

Response: We appreciate the reviewer's feedback for bringing this important issue to our attention. We agree that providing concise yet meaningful explanations for the results presented in Figures 4-6 is important for readers who may not have a deep understanding of immunopeptidomics. We revised the manuscript to reiterate the insights near the results so that the findings are clear and accessible to a wider audience.

We added the brief interpretations below to the Results section in relation to figures 4 and 6.

In relation to Fig.4 panel I: Our results suggest that the presence of glycans on MAPs does not appear to significantly affect the binding interactions between HLA molecules and peptides, as evidenced by the high similarity in HLA-binding motifs between glycosylated and non-glycosylated peptides. However, it should be noted that this conclusion is limited to glycans that successfully bind to HLA molecules. Hence, further studies utilizing HLA glyco-enriched data and larger datasets are required to gain a more detailed understanding of the potential impact of glycans on HLA molecule-peptide interactions.

The new changes can be found at lines 261 -267 in the revised manuscript.

In relation to Fig. 4 panel II and III: These findings suggest that the position of glycosylation within the HLA-binding core is not constant but varies depending on the specific HLA allele. The DPA102:01/02-DPB104:01 allele is more likely to have glycosylation within the HLA-binding core, whereas other alleles exhibit a higher likelihood of glycosylation being situated upstream or downstream.

The new changes can be found at lines 280 - 284 in the revised manuscript.

In relation to Fig. 6: Unlike the DQ and DR alleles, glycosylation within the HLA-binding core

appears to be associated with the DPA HLA gene. We speculate that this could be related to a DPA allele preference for binding truncated glyco-peptides (smaller carbohydrate moiety), causing less hindrance in the pocket. However, we did not observe a significant difference in Dalton mass between the glycans located within and outside the HLA-binding core.

The new changes can be found at lines 315 - 320 in the revised manuscript.

Comment 8 (major point): This workflow identified more than 1,000 new glyco sites, which is orders of magnitude higher number of IDs. But general analysis on this newly identified sites is missing. The authors focused on analysis of glycopeptides in the context of HLA presentation. A general analysis on these 1,000 IDs demonstrating that most of them are not all false positives should be provided, at least.

Response:

We would like to clarify that we are not claiming the identification of more than 1000 new glycosylation sites, most of these sites have been previously identified by glycoproteomics experiments. We point out in Supplementary figure 1 that 78.8% of the glycosylation sites identified in our study have already been identified and reported in GlyGen from glycoproteomics studies. Of the remaining 21% of the glycosites identified, 15.6% were in known glycoproteins, leaving only 59 glycosylation sites located on proteins that had not been reported to be glycosylated in GlyGen. Glycoproteomics and immunoproteomics workflows enrich dramatically different sections of the proteome; therefore, differences in the identified glycosites are expected. In our view, the fact that nearly 80% of our identified glycosites have been previously annotated and that 95% occur in previously identified glycoproteins provide strong evidence that the sites we observed are not all false positives.

We would also like to emphasize that we took great care to minimize the false positives in our results. Our workflow was optimized to control the false discovery rate (FDR) at 1%, and we performed extensive validation to ensure that our results were accurate, including comparison to glycosylation sites previously identified by proteomics, HLA-binding motif analysis, and HLA-binding prediction.

We added the paragraph below for clarity:

Overall, 78.8% of the detected glycosylation sites were consistent with those reported in GlyGen in previous glycoproteomics studies. Of the remaining 21%, 15.6% were detected in well-known glycoproteins, whereas only 59 were found in proteins that were not previously known to be glycosylated in GlyGen. This disparity is anticipated due to the variations in the workflows used in glycoproteomics and immunoproteomics, which target distinct parts of the proteome. Despite this, the high concurrence with previously annotated sites (80%) and known glycoproteins (95%) suggests that the identified glycosylation sites are reliable.

The new changes can be found at lines 172 - 178 in the revised manuscript.

Comment 9 (Minor points): In Fig 1, the regular search results are in grey color. Does this imply this result is not used in the whole analysis? As far as I understood, they are used in the following analyses, like Fig 4. Panel i. In the legend or in text, please specify how the results are used briefly.

Response: We apologize for any confusion that may have been caused by the use of the gray color in Fig 1. The results of the 'regular search' were used in the following analysis, and we modified its color to match the rest of the figure.

Comment 10 (Minor points): Related to Fig3. F, a quantitative analysis (like in Fig.2 F) on HLA-II and proteome exclusive glycoproteins should be provided to justify "The whole proteome glyco search likely captures glycopeptides from the most abundant glycoproteins, as the experiment was performed without any glycopeptide enrichment, whereas the immunopeptide datasets presumably capture MAPs with much less dependence on overall protein abundance."

Response: We thank the reviewer for this prompt which made us realize that the earlier statement regarding the difference in glycoproteins between MHC-associated peptides and the whole proteome due to the enrichment step may not convey the meaning we intended. During an immunopeptidomic experiment, the immunoprecipitation step captures the HLA molecules themselves using an antibody, which is then followed by the elution of the HLA-associated peptides. Therefore, any difference in glycosylation between MHC-associated peptides and the proteome is likely due to the different peptides sampled by enriching MAPs vs collecting the entire proteome from a cell lysate. The comparison between glycosylated class I and class II MHC-associated peptides versus the proteome is still intriguing. Based on the literature and our own reasoning, we hypothesized that the class II system samples peptides from the extracellular compartment, which is more likely to be rich in glycosylated proteins than the cytosol. Hence, the observed increase in glycosylation of class II-associated peptides may be attributed to the fact that the HLA processing and presentation specifically targets a subset of the proteome that likely exhibits a higher degree of glycosylation.

On the other hand, it has been suggested that proteins in the cytosol are stripped from glycans by N-glycanase before entering the cylindrical-shaped proteasome. Therefore, the depletion of glycosylation in class I-associated peptides may be due to steric hindrance posed by glycans when proteins enter the proteasome.

We added the following paragraph to our results section:

The low levels of HLA class I N-glycosylation can be explained by the fact that N-linked glycans are removed by N-glycanase before cytosolic proteins enter the cylindrical proteasome (Werdelin et al. 2002, Kario et al. 2008). In contrast, N-glycans are known to withstand MHC class II antigen processing and remain attached to associated peptides, leading to alternative glycosylated products and truncated glycans in the immunopeptidome owing to lysosomal stress (Werdelin et al. 2002). This phenomenon appears to reduce steric hindrance, providing an advantageous effect, and may explain the higher rate of glycopeptide truncation⁴¹⁻⁴³. Overall, the data showed remarkable enrichment of glycosylation in HLA class II-associated peptides relative to HLA class I and the whole proteome, leading us to focus the remainder of our efforts on HLA class II-associated and glycosylated peptides.

The new changes can be found at lines 222 - 228 in the revised manuscript.

Comment 11 (Minor points): In Fig 4 Panel I, the peptides and glyco peptides are from the

normal and glyco searches in Fig 1? And Fig4. ii and iii are only for the Glyco peptides in Fig 4. i? Even if it may be obvious, it is always helpful to clarify the scope of the data that are used to generate a specific subfigure.

Response: We apologize for not clearly indicating the scope of the data used to generate Fig 4. To clarify, **Panel I** includes both regular and glyco-search peptides while **Panels II** and **III** show only the glycopeptides. We added this information to the figure legend to ensure that the scope of the data was clear. Please find the revised Fig. 4 legend below.

We added the sentence below to the legend of figure 4.

It is worth noting that Panel I includes both regular and glyco-search peptides, whereas Panels II and III show only glycopeptides from the union of all replicates per allele.

The new changes can be found at lines 729 - 731 in the revised manuscript.

Comment 12 (Minor points): Likewise, Fig 4 ii are the peptides the ones jointly found in replicates in Fig 4 i? Or the union? From Fig 4 i, it seems that each allele has many replicates. But again the scope of the data used to generate figures is not clear.

Response: To clarify, Fig. 4 Panel II displays the peptides found in all replicates of Panel I (the union indeed). Panel I shows that each allele has many replicates, and Panel II is based on the peptides found in all replicates. We added this information to the figure legend to ensure that the scope of the data is clear.

We added the sentence below to the legend of figure 4.

It is worth noting that Panel I includes both regular and glyco-search peptides, whereas Panels II and III show only glycopeptides from the union of all replicates per allele.

The new changes can be found at lines 729 - 731 in the revised manuscript.

Comment 13 (Minor points): In Fig4. iii, for the first DPA allele, glycosylation and HLA binding seem to have a strong connection while for others not. Could there be any relevant interpretation for this localizations? The results section corresponding to this figure just ends with the observation and thus it is hard to get the take home message from this part.

Response: We appreciate the reviewer's observation regarding the strong connection between glycosylation and its location within the HLA-binding core of the first DPA allele. This observation appears to be associated with most DPA alleles in our dataset, unlike DQ and DR alleles. In the Discussion section, We speculate that this could be related to a DPA allele preference for binding truncated glyco-peptides (smaller carbohydrate moiety), causing less hindrance in the pocket. However, we did not observe a significant difference in Dalton mass between the glycans located within and outside the HLA-binding core. Another possible explanation could be attributed to a bias introduced by most DPA-binding motifs when it comes to peptides containing sequons, where fulfilling both constraints favors those located within the HLA-binding core.

We agree that the results section of the manuscript should provide a clearer take-home message for this figure, and we made the appropriate changes in the revised version. As

mentioned in our response to comment 7, we added the following paragraph to the Results section:

Unlike the DQ and DR alleles, glycosylation within the HLA-binding core appears to be associated with the DPA HLA gene. We speculate that this could be related to a DPA allele preference for binding truncated glyco-peptides (smaller carbohydrate moiety), causing less hindrance in the pocket. However, we did not observe a significant difference in Dalton mass between the glycans located within and outside the HLA-binding core.

The new changes can be found at lines 315 - 320 in the revised manuscript.

Comment 14 (Minor points): Fig. 4b panel I. DQ rep5 Glycol peptide percentage is shown to be zero. Is this an error in figure? And what is the interpretation for the similar percentages?

Response: We appreciate the feedback from the review shown in Fig. 4. b Panel I. The figure shows that there were no glycopeptides in the 5th replicate of the DQ allele. We believe that this is mostly due to the stochastic nature of data-dependent acquisition mass spectrometry, which in most cases does not produce highly replicable findings at the peptide level. In combination with the generally low percentage of glycopeptides detected (<5%) compared to the non-glycosylated immunopeptidome, the lack of glycopeptide identification does not infer their complete absence from the replicate.

We added the following paragraph to the Results section.

It is worth noting that the lack of glycopeptides in the 5th DQ allele replicate is most likely due to the stochastic nature of mass spectrometry at the peptide level. Since the percentage of glycopeptides identified in unenriched data is generally low (<5%), the lack of identification in a replicate should not be taken as evidence of their complete absence.

The new changes can be found at lines 256 - 259 in the revised manuscript.

Comment 15 (Minor points): In the discussion section, the authors mentioned “interestingly, we observed no difference in binding motif predictions with glycopeptides compared to non-glycopeptides, despite some peptides containing glycans within the binding core.” Is it interesting because this observation is against a well known prediction? If so, could there be relevant interpretation on this? The discussion section could be supplemented with more data interpretation as well.

Response: The lack of differences in HLA-binding motifs between glycopeptides and non-glycopeptides is an interesting finding that warrants further investigation. We hypothesized that the binding of glycosylated peptides to HLA molecules could either fit into known HLA-binding motifs or require glycosylation-specific motifs. Our findings suggest that glycosylation uses the same HLA-binding motifs as non-glycosylated motifs, which could indicate that the interactions inside the binding pocket are not significantly different when a carbohydrate moiety is present.

We added the following paragraph to the Discussion section:

While our current evidence suggests that glycosylation does not affect HLA-binding cores, a

more complete understanding of the potential impact of glycopeptides on HLA molecule-peptide interactions would benefit from the availability of HLA glyco-enriched data, an experimental technique that remains challenging and has yet to be accomplished. Moreover, larger datasets could facilitate the identification of altered HLA-binding cores above the level of background noise. Therefore, it is possible that our analysis did not account for glycopeptides that could modify peptide interactions with HLA molecules, highlighting the need for further research in this area.

The new changes can be found at lines 350 - 357 in the revised manuscript.

Comment 16 (Minor points): Fig. 5 and 6 also lack biological interpretation.

Response: We agree that it is important to provide biological insights and to interpret our findings in the context of previous research. We extend our discussion to include biological insights with respect to the biology of MHC II and glycosylation.

We added the paragraphs below to the Description section.

It has been suggested that N-linked glycans must be removed by N-glycanase before cytosolic proteins enter the cylindrical-shaped proteasome (Werdelin et al. 2002, Kario et al. 2008). This may explain our observation of the lack of HLA class I N-glycosylation, considering that the major source of MHC class I proteins is the cytosol. In contrast, antigen-presenting cells take up glycoproteins via endocytosis and transport them to lysosomal compartments, where proteolytic enzymes with low pH fragment them into peptides. These peptides bind to empty MHC class II molecules, forming stable MHC-peptide complexes. Studies have shown that N-glycans withstand this process and remain attached to class II MAPs (Petersen et al., 2009; Purcell et al. 2008; Werdelin et al. 2002).

Glycosylation plays a crucial role in antigen processing, as it can affect proteolysis by obstructing proteases through steric hindrance. Our study revealed variations in glycan types across HLA groups (DP, DR, and DQ), with DP alleles exhibiting an enriched presence of truncated glycans, and DR alleles demonstrating a higher mannose content. We also found that DP alleles have a higher proportion of glycans within the binding core (57%), compared to DQ alleles (30%) and DR alleles (13%). Since MHC class II processing can alter the glycan segments and result in alternative glycosylated products (Werdelin et al. 2002). Glycan truncation may provide an advantageous effect by reducing steric hindrance, leading to the higher occurrence of truncated glycans in the class II immunopeptidome than in the proteome. This may also explain the higher rate of glycopeptide truncation associated with DP alleles, as 57% of these were located within the HLA-binding core. Although glycans enhance the likelihood of presentation by making peptides more resistant to proteolysis, larger and complex glycan structures are not well recognized by T cells, as observed in previous research (Werdelin et al. 2002). It is worth noting that this analysis may not have accounted for the immune evasion that glycosylation could provide, highlighting the need for further exploration. Moreover, future research should investigate whether the correlation between smaller glycans and their presence within the HLA-binding core is solely due to size restrictions or whether it reflects other processing of MAPs for presentation.

The new changes can be found at lines 357 - 382 in the revised manuscript.

Reviewer 3

Comment 17: This manuscript performed glycosylation analyses of MAPs from 8 publicly available studies and created HLA-Glyco, a resource of HLA N-glycopeptides. Identifying glycosylated MAPs is surely important for the exploration of additional source of specificities targeted by immune responses to tumors or pathogens. While I see a potential resource aspect of the paper, unfortunately I do not think that the technical advances and findings presented have a sufficiently significant. Everything they seem to be doing is analyzing the public data with MSFragger-Glyco.

The authors stated that they developed a novel computational glyco-immunopeptidomics workflow and used one figure (Figure 1) to describe the workflow, but they indicate nothing specific to this: in what way is this more advanced than MSFragger-Glyco? It seems that the research is primarily the use of MSFragger-Glyco. Optimizing the FDR control may be not fully representative of progress, as glycopeptide FDR control has already been reported by MSFragger-Glyco (Mol. Cell. Proteomics 2022, 21, 100205), and also by several other software tools, such as GP Finder (although only for simple samples), pGlyco3, Glyco-Decipher, et al. I think they should clearly point out that if it is just an application research or there is the novelty of their glyco-immunopeptidomics workflow, as well as the improvements and differences from previous software. Benchmark with similar software would be convincing.

Response: We thank the reviewer for the valuable feedback and input.

We would like to clarify that although MSFragger-Glyco is one of the tools we use, our study goes beyond conventional (glyco-enriched, trypsin-digested) glycoproteomics data. Our glyco-immunopeptidomics workflow combines various tools and methods that were fine-tuned and integrated for the first time to enable N-glycosylation immunopeptidomic analyses. There are certainly novel methodological developments that uniquely address the challenges of glyco-immunopeptidomics, and we would like to highlight two of these developments below.

- In order to address the challenge of achieving precise FDR control in datasets that lack glycopeptide enrichment, we have introduced a novel glycopeptide-group FDR method. This method is exclusively reported in our study and is not currently available in any other glycopeptide search tools.
- The combination of peptide and glycan-level FDR control, while available in other software tools that support enzymatic glycoproteomics searches, is not possible in combination with nonspecific search except in our HLA-glyco workflow. Given the enormous search space of glyco-immunopeptides, this combination is critical to accurate glycopeptide identification without manual validation of spectra.

In response to the reviewer's feedback, we have rephrased the workflow description in the Results section to provide a clearer understanding of the advances made in this study. We invite the reviewers to take a look at the revised section and provide any further suggestions or feedback they may have.

Benchmarking against other available software. We agree that a clear demonstration of the improvements and differences from previous methods would be beneficial. However, none of the tools mentioned by the reviewer support nonspecific (non-enzymatic) searches required for immunopeptidomic analysis, including pGlyco3, Glyco-Decipher, and StrucGP. Although

the commercial ByonicTM software supports non-specific searches, when we previously reached out to the Byonic sales team we were not able to obtain a license for the software, and thus are unable to provide a comparison. We suspect that such an effort would be futile for a dataset of this size anyway, given the slow search of Byonic relative to MSFragger, even for fully enzymatic glycopeptide searches. Although MetaMorpheus supports non-specific searches through its glycopeptide search module, our benchmark yielded no glycopeptides in non-enriched datasets. This outcome suggests that MetaMorpheus, at least in our hands, may not be suitable for nonspecific searches in datasets without glycosylation enrichment (in either HLA or whole proteome data). Thus, we believe that it is not possible to perform this analysis with any other existing software tools and have emphasized this unique capability in the revised text.

We added the paragraph below to the Discussion section of the manuscript.

This study presents a novel methodology to analyze HLA N-glycopeptides using MSFragger-Glyco. Specifically, we have developed a unique glycan-group FDR method and fine-tuned a glyco-immunopeptidomics analytical workflow. These improvements enabled, for the first time, effective interrogation of non-enzymatic and non-enriched glycosylation data at a large-scale.

The new changes can be found at lines 327 - 331 in the revised manuscript.

Comment 18: The authors used the public MS data conducted without any glycopeptide enrichment. I think the identification will largely depend on the overall abundance, and the large amount of nonglycopeptides will interfere the depth and sensitivity of glycopetide identification. Why not specially perform experiments to enrich HLA glycopeptides?

Response: Immunopeptidomics requires an immense quantity of starting material (approximately 10^8 to 10^9 cells) in regular settings to obtain reliable results. With this constraint in mind, adding another layer of complexity for glycosylation enrichment is likely to make the wet-lab protocol extremely difficult. To the best of our knowledge, we are not aware of any enriched glyco-immunopeptidomics datasets.

We added the paragraphs below to the introduction.

The large-scale analysis of glycosylated MHC-associated peptides (MAPs) presents significant challenges, primarily due to the enormous search space of glycosylated non-enzymatic peptides. To address these challenges, our recent developments in improving search speed²⁰ (MSFragger) and addressing the complexity of glycosylation²¹ (MSFragger-Glyco) have proven to be beneficial. However, it is worth noting that currently, glycosylated MAPs are queried from non-enriched data, which poses a challenge as available false discovery rate (FDR) strategies are not suitable for this type of data. In light of this, we leveraged these advances to optimize a workflow for HLA glyco searches and developed an improved FDR strategy specifically for glycosylated MAPs obtained from non-enriched data.

The new changes can be found at lines 58 - 67 in the revised manuscript.

Comment 19: Interpretation of glycopeptides from proteome or immunopeptide datasets requires high search sensitivity, speed and precision. The authors analyzed 2,000 LC-MS/MS

runs and found 3409 class II N-glycosylated MAPs. I am curious about the search time and space. In addition, the depth and comprehensiveness of the identification will greatly influence the subsequent analyses and conclusions. Thus, to fully mine these public data, the comprehensive comparisons of the identifies from different MS data and widely-used software tools are highly suggested.

Response: The analysis of the eight datasets took approximately five days to complete on a server equipped with 28 cores and 512 GB of RAM. One specific dataset, consisting of 720 individual LC-MS runs, took 42.3 hours to process, equating to approximately 3.5 minutes per LC-MS file. Of the total processing time, 80% was devoted to the MSFragger search, 9% to validation and FDR filtering, and 11% to quantitation. Although the reviewer's suggestion to compare the results with other software tools is valuable, it should be noted that only HLA-Glyco is capable of performing large-scale glyco-immunopeptidomics searches. Additionally, glycopeptides are often suppressed in MS analysis when analyzed alongside non-glycopeptides, resulting in low proportions of glycopeptides unless glycopeptide enrichment is performed. As shown in Figure 2, our use of HLA-Glyco allowed us to identify significantly more glycopeptide spectra in HLA class II samples than in whole proteome samples of the same input material.

We added the paragraph below to the Discussion section of the manuscript.

Our pipeline is broadly accessible and can be executed on typical local workstation computers provided sufficient RAM is available (a minimum of 64 GB of RAM is required). The analysis of the eight HLA class II datasets took approximately five days to complete on a server equipped with 28 cores and 512 GB of RAM. As an example, one dataset consisting of 720 individual LC-MS runs took 42.3 hours to process, equating to approximately 3.5 minutes per LC-MS file. Of the total processing time, 80% was devoted to MSFragger search, 9% to validation and FDR filtering, and 11% for quantitation. These results demonstrate the feasibility of our approach for large-scale, comprehensive analysis of glycosylation patterns in non-enriched datasets, which is critical for understanding the glycoproteome in biological samples.

The new changes can be found at lines 408 -414 in the revised manuscript.

Comment 20: They performed insufficient depth of the result mining and analysis, from which, we could get very limited information. For example, one of the main findings is that they find a higher percentage of truncated glycans (68%) in the HLA-II sample. While no other exploration was made. The quantitative and heterogeneity analysis of these glycans are necessary for us to characterize these HLA N-glycopeptides. For HLA-glyco data, the authors mainly focus on the N-glycosylation-driven HLA motif analysis. However, glycan-related motif information was ignored. Besides, conclusions, such as sample specific signatures, unique glyco-antigens, are also hard to get from the current analyses.

Response: We appreciate the reviewers' suggestion for additional analyses and interpretations of the presented resource. In this study, we focused on developing a resource and providing a reproducible glyco-immunopeptidomic pipeline. We have provided all the necessary tools and data for further research in this field. Additionally, we have revised the manuscript to include additional discussion and interpretation of biological data in response to the feedback from Reviewer 2 (see our responses to Reviewer 2). Although the antigenicity of

the peptides is not within the scope of this study, we hope that the data provided will assist future research in assessing the immunogenicity of HLA-associated peptides.

We have extensively revised the discussion section to ensure clarity and provide greater insight into our research findings. We would like to invite the reviewers to review our revised discussion section for their feedback and suggestions. Moreover, the motifs were well addressed in the manuscript (see **Supplementary Figure 2**).

We have added below the most interesting additions in this particular context for the convenience of the reviewer.

It has been suggested that N-linked glycans must be removed by N-glycanase before cytosolic proteins enter the cylindrical-shaped proteasome (Werdelin et al. 2002, Kario et al. 2008). This may explain our observation of the lack of HLA class I N-glycosylation, considering that the major source of MHC class I proteins is the cytosol. In contrast, antigen-presenting cells take up glycoproteins via endocytosis and transport them to lysosomal compartments, where proteolytic enzymes with low pH fragment them into peptides. These peptides bind to empty MHC class II molecules, forming stable MHC-peptide complexes. Studies have shown that N-glycans withstand this process and remain attached to class II MAPs (Petersen et al., 2009; Purcell et al. 2008; Werdelin et al. 2002).

Glycosylation plays a crucial role in antigen processing, as it can affect proteolysis by obstructing proteases through steric hindrance. Our study revealed variations in glycan types across HLA groups (DP, DR, and DQ), with DP alleles exhibiting an enriched presence of truncated glycans, and DR alleles demonstrating a higher mannose content. We also found that DP alleles have a higher proportion of glycans within the binding core (57%), compared to DQ alleles (30%) and DR alleles (13%). Since MHC class II processing can alter the glycan segments and result in alternative glycosylated products (Werdelin et al. 2002). Glycan truncation may provide an advantageous effect by reducing steric hindrance, leading to the higher occurrence of truncated glycans in the class II immunopeptidome than in the proteome. This may also explain the higher rate of glycopeptide truncation associated with DP alleles, as 57% of these were located within the HLA-binding core. Although glycans enhance the likelihood of presentation by making peptides more resistant to proteolysis, larger and complex glycan structures are not well recognized by T cells, as observed in previous research (Werdelin et al. 2002). It is worth noting that this analysis may not have accounted for the immune evasion that glycosylation could provide, highlighting the need for further exploration. Moreover, future research should investigate whether the correlation between smaller glycans and their presence within the HLA-binding core is solely due to size restrictions or whether it reflects other processing of MAPs for presentation.

The new changes can be found at lines 357 - 382 in the revised manuscript.

Comment 21: The O-glycosylated MAPs are also of potentials interest as described in the paper. In the discussion part, the authors should objectively discuss the difficulties and the key points of O-glycosylated MAPs identification rather than simply saying that “in principle, be studied by our method”, since there is no data to support this conclusion. I do not think O-glycosylated MAPs can be easily interpreted from these public MS data using the workflow presented here.

Response: We agree with the reviewer that O-glycosylated MHC-associated peptides represent an exciting and challenging frontier. Our current workflows for O-glycopeptide data can use all the innovations described here and thus, in theory, are capable of the same analysis. Particularly now that we have added the O-Pair method for O-glycan localization to FragPipe (not yet published). However, the major limitation, and perhaps what the reviewer is referring to, is the relative scarcity of electron-based activation (e.g., ETD) based datasets required to confidently localize O-glycans. We note that such datasets exist, and our workflow could be used to search for O-glycosylated MHC-associated peptides, although on a much smaller scale than N-glycosylation, given the smaller number of available datasets. We have revised this statement in the text to clarify these limitations and noted the need for electron-based activation to employ our workflow for O-glycosylated MAPs.

We added the paragraph below to the revised manuscript.

With our aim to support tumor antigen discovery and broaden potential targets, we developed a workflow to study N-glycosylated MAPs. Moreover, we have recently added an MSFragger Glyco-compatible version of O-Pair (Marino et al. 2015, Lu et al. 2020) to FragPipe, enabling confident annotation of O-glycosylated MAPs. However, the scarcity of electron-based activation datasets required to locate O-glycans currently limits its large-scale application. We anticipate that the findings of this work will spark increasing interest in glycosylation studies within the field of immunopeptidomics.

The new changes can be found at lines 409 - 415 in the revised manuscript.

Reviewers' Comments:

Reviewer #2:

Remarks to the Author:

The authors addressed all the points clearly except for one point. The only point I am not clear about is again the FDR issue. I agree that the number of decoys and targets filtered by the sequon and reversed sequon are expected to be the same. But how about match scores? Do spectrum-target and spectrum-decoy have similar scores? I think the glyco spectra will have signature for the sequon. Then after filtration, the match score between the spectra with sequon and target sequence with sequon would not be higher than the spectra vs. decoy without sequon?

The crucial assumption of target decoy is this: false positives in target and decoy hits should have the same SCORE distributions. Not just number of sequences. I checked the paper (<https://www.nature.com/articles/s41592-020-0967-9>) but this point is not discussed anywhere. I believe this effect is minor but should be addressed to make sure that it is indeed minor. If the spectra with sequon signatures are not expected to increase the match score against the peptides with sequons, it should be clarified somewhere. Otherwise comparison of false target and decoy distributions would be helpful.

Reviewer #3:

Remarks to the Author:

The manuscript presents a computational workflow for glycopeptide identification from MS-based immunopeptidome data. The central idea of the paper is that it gets through the glycopeptide identification pipeline in immunopeptidome data and therefore provides a source of HLA-Glyco data. The authors have revised the manuscript, addressing main concerns with adding more discussions and improving the clarity of their explanations. However, some issues remain unresolved.

One important aspect of the manuscript is the authors' proposed FDR control method for filtering glycopeptides from the large-scale nonglycopeptides in the glycopeptide non-enriched MS data. Although the authors emphasize the efficiency of their method, it lacks proper validation through benchmark experimental data. To address this, the authors could merge enriched-glycopeptide MS data from one species with nonglycopeptide data from another species to create a combined dataset for searching target glycopeptides. This would demonstrate the reliability of their FDR control method.

Another concern is the authors' claim of introducing a fast computational workflow. However, as clarified in their response, the analysis of the eight datasets took approximately five days to complete, which may not be considered fast in some contexts. Therefore, the authors should reconsider their assertion regarding the speed of their workflow.

Response to reviewers' comments on the manuscript "Unraveling the glycosylated immunopeptidome with HLA-Glyco" (Reference number NCOMMS-22-45486A).

Please find below our responses to all the comments received from the reviewers. Where appropriate, the location of the associated edit is indicated by line number (L) in the revised manuscript.

The document follows the color code below to facilitate this process:

- Reviewer's comment.
- Response to the comment.
- Location of the associated edits in the revised manuscript and/or revised supplementary notes in docx format.

Reviewer 2

Comment 1: The authors addressed all the points clearly except for one point. The only point I am not clear about is again the FDR issue. I agree that the number of decoys and targets filtered by the sequon and reversed sequon are expected to be the same. But how about match scores? Do spectrum-target and spectrum-decoy have similar scores? I think the glyco spectra will have signature for the sequon. Then after filtration, the match score between the spectra with sequon and target sequence with sequon would not be higher than the spectra vs. decoy without sequon?

The crucial assumption of target decoy is this: false positives in target and decoy hits should have the same SCORE distributions. Not just number of sequences. I checked the paper (<https://www.nature.com/articles/s41592-020-0967-9>) but this point is not discussed anywhere. I believe this effect is minor but should be addressed to make sure that it is indeed minor. If the spectra with sequon signatures are not expected to increase the match score against the peptides with sequons, it should be clarified somewhere. Otherwise comparison of false target and decoy distributions would be helpful.

Response: We have conducted an analysis and confirmed that the spectra of peptides, both with and without sequons, exhibit identical score distributions for decoy and low-scoring (false) targets, as we anticipated. This finding aligns with our expectations since the presence of a sequon should not influence the probability of randomly matching fragment ions to peaks in a spectrum. For further reference, we have included the score distributions in the supporting information.

The hyperscore distributions for target and decoy peptides with and without sequons have been added as Supplementary Figure 3.

The manuscript now contains the following revised statement at lines 440 to 441: We confirmed that score distributions for decoys and false targets for peptides with and without sequons are highly similar (Supplementary Figure 3).

Reviewer 3

Comment 2: The manuscript presents a computational workflow for glycopeptide identification from MS-based immunopeptidome data. The central idea of the paper is that it gets through the glycopeptide identification pipeline in immunopeptidome data and therefore provides a source of HLA-Glyco data. The authors have revised the manuscript, addressing main concerns with adding more discussions and improving the clarity of their explanations. However, some issues remain unresolved.

One important aspect of the manuscript is the authors' proposed FDR control method for filtering glycopeptides from the large-scale nonglycopeptides in the glycopeptide non-enriched MS data. Although the authors emphasize the efficiency of their method, it lacks proper validation through benchmark experimental data. To address this, the authors could merge enriched-glycopeptide MS data from one species with nonglycopeptide data from another species to create a combined dataset for searching target glycopeptides. This would demonstrate the reliability of their FDR control method.

Response: We appreciate the reviewer's positive evaluation of our manuscript revision and their recognition of the central idea behind our computational workflow for glycopeptide identification.

To demonstrate the reliability of our FDR control, we conducted two entrapment searches. As suggested by the reviewer, the first search involved merging enriched glycopeptide raw data from another species with an equal amount of human HLA data. Using a human database only, we performed the HLA-glyco search on 3 raw files of yeast N-glycopeptide enriched data alongside 3 human HLA raw files. After applying our FDR filtering, no glycoPSMs were detected in the yeast raw files, confirming that our method effectively controls the rate of matching to glycopeptide spectra even in the presence of enriched glycopeptide spectra from another species.

In addition, we performed another entrapment search using a more typical strategy. This involved using an entrapment protein database from another species to search the HLA data using our workflow. Specifically, we searched one of the eight HLA datasets (131 raw files) by combining the Human and Arabidopsis thaliana proteome. Similar to the previous search, none of the A. thaliana glycoPSMs passed our combined peptide and glycan FDR filtering, resulting in an empirical FDR of 0%. When considering all spectra, not just glycoPSMs, the entrapment rate was found to be 0.03%.

These entrapment searches demonstrate the highly stringent and effective nature of our FDR control procedures. The results of both searches have been included in the supporting information.

Both entrapment searches and results are described in **Supplementary Note 2** and **Supplementary Data 1**.

Please refer to lines 138 to 144 in the revised manuscript regarding the entrapment search results.

Comment 3: Another concern is the authors' claim of introducing a fast computational

workflow. However, as clarified in their response, the analysis of the eight datasets took approximately five days to complete, which may not be considered fast in some contexts. Therefore, the authors should reconsider their assertion regarding the speed of their workflow.

Response: The analysis took an average of 2.7 minutes per raw mass spectrometry file to complete, taking multiple days only because we processed thousands of files. In our view, this is fast given the search space. We have revised this statement to note the average search time per raw file in addition to the total time to avoid confusion.

The manuscript now contains the following revised statement in the discussion at lines 401-403: “The analysis of the eight large HLA class II datasets took 52.3 hours, or 2.7 minutes per raw file, to complete on a server equipped with 28 cores and 512 GB of RAM.”.